# Unbiased Supervised Contrastive Learning

**Carlo Alberto Barbano**[*]
University of Turin
LTCI, Telécom Paris, IP Paris

**Benoit Dufumier**
LTCI, Télécom Paris, IP Paris

**Enzo Tartaglione**
LTCI, Télécom Paris, IP Paris

**Marco Grangetto**
University of Turin

**Pietro Gori**
LTCI, Télécom Paris, IP Paris

## Abstract

Many datasets are biased, namely they contain easy-to-learn features that are highly correlated with the target class only in the dataset but not in the true underlying distribution of the data. For this reason, learning unbiased models from biased data has become a very relevant research topic in the last years. In this work, we tackle the problem of learning representations that are robust to biases. We first present a margin-based theoretical framework that allows us to clarify why recent contrastive losses (InfoNCE, SupCon, etc.) can fail when dealing with biased data. Based on that, we derive a novel formulation of the supervised contrastive loss ($\epsilon$-*SupInfoNCE*), providing more accurate control of the minimal distance between positive and negative samples. Furthermore, thanks to our theoretical framework, we also propose *FairKL*, a new debiasing regularization loss, that works well even with extremely biased data. We validate the proposed losses on standard vision datasets including CIFAR10, CIFAR100, and ImageNet, and we assess the debiasing capability of FairKL with $\epsilon$-SupInfoNCE, reaching state-of-the-art performance on a number of biased datasets, including real instances of biases "in the wild".

## 1 Introduction

Deep learning models have become the predominant tool for learning representations suited for a variety of tasks. Arguably, the most common setup for training deep neural networks in supervised classification tasks consists in minimizing the cross-entropy loss. Cross-entropy drives the model towards learning the correct label distribution for a given sample. However, it has been shown in many works that this loss can be affected by biases in the data (Alvi et al., 2018; Kim et al., 2019; Nam et al., 2020; Sagawa et al., 2019; Tartaglione et al., 2021; Torralba et al., 2011) or suffer by noise and corruption in the labels Elsayed et al. (2018); Graf et al. (2021). In fact, in the latest years, it has become increasingly evident how neural networks tend to rely on simple patterns in the data (Geirhos et al., 2019; Li et al., 2021). As deep neural networks grow in size and complexity, guaranteeing that they do not learn spurious elements in the training set is becoming a pressuring issue to tackle. It is indeed a known fact that most of the commonly-used datasets are biased (Torralba et al., 2011) and that this affects the learned models (Tommasi et al., 2017). In particular, when the biases correlate very well with the target task, it is hard to obtain predictions that are independent of the biases. This can happen, e.g., in presence of selection biases in the data. Furthermore, if the bias is easy to learn (e.g. a simple pattern or color), we will most likely obtain a biased model, whose predictions majorly rely on these spurious attributes and not on the true, generalizable, and discriminative features. Learning fair and robust representations of the underlying samples, especially when dealing with highly-biased data, is the main objective of this work. Contrastive learning has recently gained attention for this purpose, showing superior robustness to cross-entropy Graf et al. (2021). For this reason, in this work, we adopt a metric learning approach for supervised representation learning. Based on that, we provide a unified framework to analyze and compare existing formulations of contrastive losses[1] such as the InfoNCE loss (Chen et al., 2020; Oord et al.,

---

[*]Corresponding author: `carlo.barbano@unito.it`
[1]We refer to any contrastive loss and not necessarily to losses based on pairs of samples as in (Sohn, 2016).

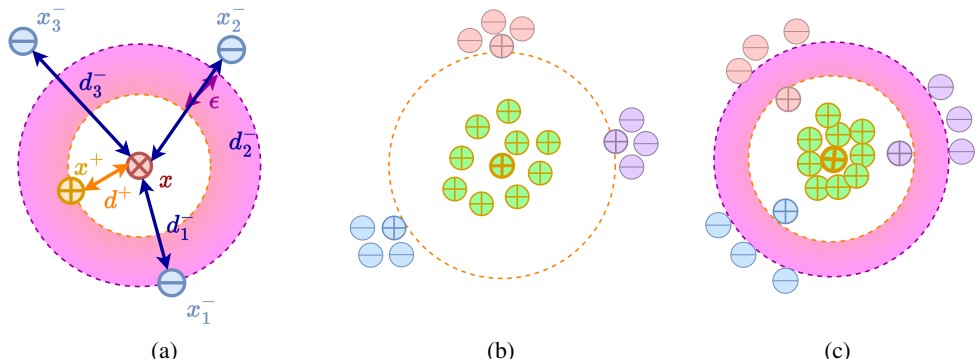

Figure 1: With $\epsilon$-SupInfoNCE (a) we aim at increasing the minimal margin $\epsilon$, between the distance $d^+$ of a positive sample $x^+$ (+ symbol inside) from an anchor $x$ and the distance $d^-$ of the closest negative sample $x^-$ (− symbol inside). By increasing the margin, we can achieve a better separation between positive and negative samples. We show two different scenarios without margin (b) and with margin (c). Filling colors of datapoints represent different biases. We observe that, without imposing a margin, biased clusters might appear containing both positive and negative samples (b). This issue can be mitigated by increasing the $\epsilon$ margin (c).

2019), the InfoL1O loss (Poole et al., 2019) and the SupCon loss (Khosla et al., 2020). Furthermore, we also propose a new supervised contrastive loss that can be seen as the simplest extension of the InfoNCE loss (Chen et al., 2020; Oord et al., 2019) to a supervised setting with multiple positives. Using the proposed metric learning approach, we can reformulate each loss as a set of contrastive, and surprisingly sometimes even non-contrastive, conditions. We show that the widely used SupCon loss is not a "straightforward" extension of the InfoNCE loss since it actually contains a set of "latent" non-contrastive constraints. Our analysis results in an in-depth understanding of the different loss functions, fully explaining their behavior from a metric point of view. Furthermore, by leveraging the proposed metric learning approach, we explore the issue of biased learning. We outline the limitations of the studied contrastive loss functions when dealing with biased data, even if the loss on the training set is apparently minimized. By analyzing such cases, we provide a more formal characterization of bias. This eventually allows us to derive a new set of regularization constraints for debiasing that is general and can be added to any contrastive or non-contrastive loss. Our contributions are summarized below:

1. We introduce a simple but powerful theoretical framework for supervised representation learning, from which we derive different contrastive loss functions. We show how existing contrastive losses can be expressed within our framework, providing a uniform understanding of the different formulations. We derive a generalized form of the SupCon loss ($\epsilon$-SupCon), propose a novel loss $\epsilon$-*SupInfoNCE*, and demonstrate empirically its effectiveness;

2. We provide a more formal definition of bias, thanks to the proposed metric learning approach, which is based on the distances among representations, This allows us to derive a new set of effective debiasing regularization constraints, which we call *FairKL*. We also analyze, theoretically and empirically, the debiasing power of the different contrastive losses, comparing $\epsilon$-SupInfoNCE and SupCon.

## 2 RELATED WORKS

Our work is related to the literature in contrastive learning, metric learning, fairness, and debiasing.
**Contrastive Learning** Many different contrastive losses and frameworks have been proposed (Chen et al., 2020; Khosla et al., 2020; Oord et al., 2019; Poole et al., 2019). Supervised contrastive learning approaches aim at pulling representations of the same class close together while repelling representations of different classes apart from each other. It has been shown that, in a supervised setting, this kind of optimization can yield better results than standard cross-entropy, and can be more robust against label corruption (Khosla et al., 2020; Graf et al., 2021). Related to contrastive learning, we

can also find methods such as triplet losses (Chopra et al., 2005; Hermans et al., 2017) and based on distance metrics Schroff et al. (2015); Weinberger et al. (2006). The latter are the most relevant for this work, as we propose a metric learning approach for supervised representation learning.

**Debiasing** Addressing the issue of biased data and how it affects generalization in neural networks has been the subject of numerous works. Some approaches in this direction include the use of different data sources in order to mitigate biases (Gupta et al., 2018) and data clean-up thanks to the use of a GAN (Sattigeri et al., 2018; Xu et al., 2018). However, they share some major limitations due to the complexity of working directly on the data. In the debiasing related literature, we can most often find approaches based on ensembling or adversarial setups, and regularization terms that aim at obtaining an *unbiased* model using *biased* data. The typical adversarial approach is represented by BlindEye (Alvi et al., 2018). They employ an explicit bias classifier, trained on the same representation space as the target classifier, in an adversarial way, forcing the encoder to extract unbiased representations. This is also similar to Xie et al. (2017). Kim et al. (2019) use adversarial learning and gradient inversion to reach the same goal. Wang et al. (2019b) adopt an adversarial approach to remove unwanted features from intermediate representations of a neural network. All of these works share the limitations of adversarial training, which is well known for its potential training instability. Other ensembling approaches can be found in Clark et al. (2019); Wang et al. (2020), where feature independence among the different models is promoted. Bahng et al. (2020) propose ReBias, aiming at promoting independence between biased and unbiased representations with a min-max optimization. In (Lee et al., 2021) disentanglement between bias features and target features is maximized to perform the augmentation in the latent space. Nam et al. (2020) propose LfF, where a bias-capturing model is trained with a focus on easier samples (bias-aligned), while a debiased network is trained by giving more importance to the samples that the bias-capturing model struggles to discriminate. Another approach is proposed in Wang et al. (2019a) with HEX, where a neural-network-based gray-level co-occurrence matrix (Haralick et al., 1973; Lam, 1996), is employed for learning invariant representations to some bias. However, all these methods require training additional models, which in practice can be resource and time-consuming. Obtaining representations that are robust and/or invariant to some secondary attribute can also be achieved by applying constraints and regularization to the model. Using regularization terms for debiasing has gained traction also due to the typically lower complexity when compared to methods such as exembling. For example, recent works attempt to discourage the learning of certain features with the aim of data privacy (Barbano et al., 2021; Song et al., 2017) and fairness (Beutel et al., 2019). For example, Sagawa et al. (2019) propose Group-DRO, which aims at improving the model performance on the *worst-group*, defined based on prior knowledge of the bias distribution. In RUBi (Cadene et al., 2019), logits re-weighting is used to promote the independence of the predictions on the bias features. Tartaglione et al. (2021) propose EnD, which is a regularization term that aims at bringing representations of positive samples closer together in case of different biases, and pulling apart representations of negative samples sharing the same bias attributes. A similar method is presented in (Hong & Yang, 2021), where a contrastive formulation is employed to reach a similar goal. Our method belongs to this latter class of approaches, as it consists of a regularization term which can be optimized during training.

## 3 Contrastive learning: an $\epsilon$-margin point of view

Let $x \in \mathcal{X}$ be an original sample (i.e., anchor), $x_i^+$ a similar (positive) sample, $x_j^-$ a dissimilar (negative) sample and $P$ and $N$ the number of positive and negative samples respectively. Contrastive learning methods look for a parametric mapping function $f : \mathcal{X} \to \mathbb{S}^{d-1}$ that maps "semantically" similar samples close together in the representation space (a $(d\text{-}1)$-sphere) and dissimilar samples far away from each other. Once pre-trained, $f$ is fixed and its representation is evaluated on a downstream task, such as classification, through linear evaluation on a test set. In general, positive samples $x_i^+$ can be defined in different ways depending on the problem: using transformations of $x$ (unsupervised setting), samples belonging to the same class as $x$ (supervised) or with similar image attributes of $x$ (weakly-supervised). The definition of negative samples $x_j^-$ varies accordingly. Here, we focus on the supervised case, thus samples belonging to the same/different class, but the proposed framework could be easily applied to the other cases. We define $s(f(a), f(b))$ as a similarity measure (e.g., cosine similarity) between the representation of two samples $a$ and $b$. Please note that since $||f(a)||_2 = ||f(b)||_2 = 1$, using a cosine similarity is equivalent to using a L2-distance $(d(f(a), f(b)) = ||f(a) - f(b)||_2^2)$. Similarly to Chopra et al. (2005); Hadsell et al. (2006); Schroff et al. (2015); Sohn (2016); Wang et al. (2014; 2019c); Weinberger et al. (2006); Yu & Tao (2019),

we propose to use a metric learning approach which allows us to better formalize recent contrastive losses, such as InfoNCE (Chen et al., 2020; Oord et al., 2019), InfoL1O (Poole et al., 2019) and SupCon (Khosla et al., 2020), and derive new losses that better approximate the mutual information and can take into account data biases. Using an $\epsilon$-margin metric learning point of view, probably the simplest contrastive learning formulation is looking for a mapping function $f$ such that the following $\epsilon$-condition is always satisfied:

$$\underbrace{d(f(x), f(x^+))}_{d^+} - \underbrace{d(f(x), f(x_j^-))}_{d_j^-} < -\epsilon \iff \underbrace{s(f(x), f(x_j^-))}_{s_j^-} - \underbrace{s(f(x), f(x^+))}_{s^+} \leq -\epsilon \quad \forall j \quad (1)$$

where $\epsilon \geq 0$ is a margin between positive and negative samples and we consider, for now, a single positive sample.

**Derivation of InfoNCE** The constraint of Eq. 1 can be transformed in an optimization problem using, as it is common in contrastive learning, the max operator and its smooth approximation *LogSumExp* (full derivation in the Appendix A.1.1):

$$s_j^- - s^+ \leq -\epsilon \quad \forall j$$

$$\underset{f}{\arg\min} \max(-\epsilon, \{s_j^- - s^+\}_{j=1,\dots,N}) \approx \underset{f}{\arg\min} \underbrace{-\log\left(\frac{\exp(s^+)}{\exp(s^+ - \epsilon) + \sum_j \exp(s_j^-)}\right)}_{\epsilon-InfoNCE} \quad (2)$$

Here, we can notice that when $\epsilon = 0$, we retrieve the InfoNCE loss, also known as N-Pair loss (Sohn, 2016), whereas when $\epsilon \to \infty$ we obtain the InfoL1O loss. It has been shown in Poole et al. (2019) that these two losses are lower and upper bound of the Mutual Information $I(X^+, X)$ respectively:

$$\underbrace{\log \frac{\exp s^+}{\exp s^+ + \sum_j \exp s_j^-}}_{InfoNCE} \leq I(X^+, X) \leq \underbrace{\log \frac{\exp s^+}{\sum_j \exp s_j^-}}_{InfoL1O} \quad (3)$$

By using a value of $\epsilon \in [0, \infty)$, one might find a tighter approximation of $I(X^+, X)$ since the exponential function at the denominator $\exp(-\epsilon)$ monotonically decreases as $\epsilon$ increases.

**Proposed supervised loss ($\epsilon$-SupInfoNCE)** The inclusion of multiple positive samples ($s_i^+$) can lead to different formulations. Some of them can be found in the Appendix A.1.2. Here, considering a supervised setting, we propose to use the following one, that we call $\epsilon$-SupInfoNCE:

$$s_j^- - s_i^+ \leq -\epsilon \quad \forall i, j$$

$$\sum_i \max(-\epsilon, \{s_j^- - s_i^+\}_{j=1,\dots,N}) \approx \underbrace{-\sum_i \log\left(\frac{\exp(s_i^+)}{\exp(s_i^+ - \epsilon) + \sum_j \exp(s_j^-)}\right)}_{\epsilon-SupInfoNCE} \quad (4)$$

Please note that this loss could also be used in other settings, like in an unsupervised one, where positive samples could be defined as transformations of the anchor. Furthermore, even here, the $\epsilon$ value can be adjusted in the loss function, in order to increase the $\epsilon$-margin. This time, contrarily to what happens with Eq. 2 and InfoNCE, if we consider $\epsilon = 0$, we do not obtain the SupCon loss.

**Derivation of $\epsilon$-SupCon (generalized SupCon)** It's interesting to notice that Eq. 4 is similar to $\mathcal{L}_{out}^{sup}$, which is one of the two SupCon losses proposed in Khosla et al. (2020), but they differ for a sum over the positive samples at the denominator. The $\mathcal{L}_{out}^{sup}$ loss, presented as the "most straightforward way to generalize" the InfoNCE loss, actually contains another non-contrastive constraint on the positive samples: $s_t^+ - s_i^+ \leq 0 \quad \forall i, t$. Fulfilling this condition alone would force all positive samples to collapse to a single point in the representation space. However, it does not take into account negative samples. That is why we define it as a non-contrastive condition. Considering both

contrastive and non-contrastive conditions, we obtain:

$$s_j^- - s_i^+ \leq -\epsilon \quad \forall i,j \quad \text{and} \quad s_t^+ - s_i^+ \leq 0 \quad \forall i, t \neq i$$

$$\frac{1}{P}\sum_i \max(0, \{s_j^- - s_i^+ + \epsilon\}_j, \{s_t^+ - s_i^+\}_{t\neq i}) \approx \epsilon - \underbrace{\frac{1}{P}\sum_i \log\left(\frac{\exp(s_i^+)}{\sum_t \exp(s_t^+ - \epsilon) + \sum_j \exp(s_j^-)}\right)}_{\epsilon - SupCon}$$

(5)

when $\epsilon = 0$ we retrieve exactly $\mathcal{L}_{out}^{sup}$. The second loss proposed in Khosla et al. (2020), called $\mathcal{L}_{in}^{sup}$, minimizes a different contrastive problem, which is a less strict condition and probably explains the fact that this loss did not work well in practice (Khosla et al., 2020):

$$\max(s_j^-) < \max(s_i^+) \approx \log\left(\sum_j \exp(s_j^-)\right) - \log\left(\sum_i \exp(s_i^+)\right) < 0 \tag{6}$$

$$\arg\min_f \max(0, \max(s_j^-) - \max(s_i^+)) \approx -\underbrace{\log\left(\sum_i \frac{\exp(s_i^+)}{\sum_t \exp(s_t^+) + \sum_j \exp(s_j^-)}\right)}_{\mathcal{L}_{in}^{sup}} \tag{7}$$

It's easy to see that, differently from Eq. 4 and $\mathcal{L}_{out}^{sup}$, this condition is fulfilled when just *one* positive sample is more similar to the anchor than all negative samples. Similarly, another contrastive condition that should be avoided is: $\sum_j s(f(x), f(x_j^-)) - \sum_i s(f(x), f(x_i^+)) < -\epsilon$ since one would need only *one* (or few) negative samples far away from the anchor in the representation space (i.e., orthogonal) to fulfil the condition.

### 3.1 FAILURE CASE OF INFONCE: THE ISSUE OF BIASES

Satisfying the $\epsilon$-condition (1) can generally guarantee good downstream performance, however, it does not take into account the presence of biases (e.g. selection biases). A model could therefore take its decision based on certain visual features, i.e. the bias, that are correlated with the target downstream task but don't actually characterize it. This means that the same bias features would probably have a worse performance if transferred to a different dataset (e.g. different acquisition settings or image quality). Specifically, in contrastive learning, this can lead to settings where we are still able to minimize any InfoNCE-based loss (e.g. SupCon or $\epsilon$-SupInfoNCE), but with degraded classification performance (Fig. 1b). To tackle this issue, in this work, we propose the FairKL regularization technique, a set of debiasing constraints that prevent the use of the bias features within the proposed metric learning approach. In order to give a more in-depth explanation of the $\epsilon$-InfoNCE failure case, we employ the notion of *bias-aligned* and *bias-conflicting* samples as in Nam et al. (2020). In our context, a bias-aligned sample shares the same bias attribute of the anchor, while a bias-conflicting sample does not. In this work, we assume that the bias attributes are either known *a priori* or that they can be estimated using a bias-capturing model, such as in Hong & Yang (2021).

**Characterization of bias** We denote bias-aligned samples with $x^{\cdot,b}$ and bias-conflicting samples with $x^{\cdot,b'}$. Given an anchor $x$, if the bias is "strong" and easy-to-learn, a *positive bias-aligned* sample $x^{+,b}$ will probably be closer to the anchor $x$ in the representation space than a *positive bias-conflicting* sample (of course, the same reasoning can be applied for the negative samples). This is why even in the case in which the $\epsilon$-condition is satisfied and the $\epsilon$-SupInfoNCE is minimized, we could still be able to distinguish between bias-aligned and bias-conflicting samples. Hence, we say that there is a bias if we can identify an ordering on the learned representations, such as:

$$\underbrace{d(f(x), f(x_i^{+,b}))}_{d_i^{+,b}} < \underbrace{d(f(x), f(x_k^{+,b'}))}_{d_k^{+,b'}} \leq \underbrace{d(f(x), f(x_t^{-,b}))}_{d_t^{-,b}} - \epsilon < \underbrace{d(f(x), f(x_j^{-,b'}))}_{d_j^{-,b'}} - \epsilon \quad \forall i,k,t,j$$

(8)

This represents the worst-case scenario, where the ordering is total (i.e., $\forall i,k,t,j$). Of course, there can also be cases in which the bias is not as strong, and the ordering may be partial.

**FairKL regularization for debiasing** Ideally, we would enforce the conditions $d_k^{+,b'} - d_i^{+,b} = 0 \quad \forall i,k$ and $d_t^{-,b'} - d_j^{-,b} = 0 \quad \forall t,j$, meaning that every positive (resp. negative) bias-conflicting

sample should have the same distance from the anchor as any other positive (resp. negative) bias-aligned sample. However, in practice, this condition is very strict, as it would enforce uniform distance among all positive (resp. negative) samples. A more relaxed condition would instead force the distributions of distances, $\{d_k^{\cdot,b'}\}$ and $\{d_i^{\cdot,b}\}$, to be similar. Here, we propose two new debiasing constraints for both positive and negative samples using either the first moment (mean) of the distributions or the first two moments (mean and variance). Using only the average of the distributions, we obtain:

$$\frac{1}{P_a}\sum_i d_i^{+,b} - \frac{1}{P_c}\sum_k d_k^{+,b'} = 0 \iff \frac{1}{P_c}\sum_k |s_k^{+,b'}| - \frac{1}{P_a}\sum_i |s_i^{+,b}| = 0 \qquad (9)$$

where $P_a$ and $P_c$ are the number of positive bias-aligned and bias-conflicting samples, respectively[2].

Denoting the first moments with $\mu_{+,b} = \frac{1}{P_a}\sum_i d_i^{+,b}$, $\mu_{+,b'} = \frac{1}{P_c}\sum_k d_k^{+,b'}$, and the second moments of the distance distributions with $\sigma_{+,b}^2 = \frac{1}{P_a}\sum_i (d_i^{+,b} - \mu_{+,b})^2$, $\sigma_{+,b'}^2 = \frac{1}{P_c}\sum_k (d_k^{+,b'} - \mu_{+,b-})^2$, and making the hypothesis that the distance distributions follow a normal distribution, we can define a new set of debiasing constraints using, for example, the Kullback–Leibler divergence:

$$D_{KL}(\{d_i^{+,b}\}||\{d_k^{+,b'}\}) = \frac{1}{2}\left[\frac{\sigma_{+,b}^2 + (\mu_{+,b} - \mu_{+,b'})^2}{\sigma_{+,b'}^2} - \log\frac{\sigma_{+,b}^2}{\sigma_{+,b'}^2} - 1\right] = 0 \qquad (10)$$

In practice, one could also use another distribution such as the log-normal, the Jeffreys divergence ($D_{KL}(p||q) + D_{KL}(q||p)$), or a simplified version, such as the difference of the two statistics (e.g., $(\mu_{+,b} - \mu_{+,b'})^2 + (\sigma_{+,b} - \sigma_{+,b'})^2$). The proposed debiasing constrains can be easily added to any contrastive loss using the method of the Lagrange multipliers, as a regularization term $\mathcal{R}^{FairKL} = D_{KL}(\{d_i^{+,b}\}||\{d_k^{+,b'}\})$. Thus, the final loss function that we propose to minimize is:

$$\mathcal{L} = \underbrace{-\alpha\sum_i \log\left(\frac{\exp(s_i^+)}{\exp(s_i^+ - \epsilon) + \sum_j \exp(s_j^-)}\right)}_{\epsilon-SupInfoNCE} + \lambda\mathcal{R}^{FairKL} \qquad (11)$$

where $\alpha$ and $\lambda$ are positive hyperparameters.

### 3.1.1 COMPARISON WITH OTHER DEBIASING METHODS

**SupCon** It is interesting to notice that the non-contrastive conditions in Eq. 5: $s_t^+ - s_i^+ \leq 0 \quad \forall i, t \neq i$ are actually all fulfilled only when $s_i^+ = s_t^+ \quad \forall i, t \neq i$. This means that one tries to align all positive samples, regardless of their bias $b$, to a single point in the representation space. In other terms, at the optimal solution, one would also fulfill the following conditions:

$$s_i^{+,b} = s_t^{+,b}, s_i^{+,b'} = s_t^{+,b'}, s_i^{+,b} = s_t^{+,b'}, s_i^{+,b'} = s_t^{+,b} \quad \forall i, t \neq i \qquad (12)$$

Realistically, this could lead to suboptimal solutions: we argue that the optimization process would mainly focus on the easier task, namely aligning bias-aligned samples, and neglecting the bias-conflicting ones. In highly biased settings, this could lead to worse performance than $\epsilon$-SupInfoNCE. More empirical results supporting this hypothesis are presented in Appendix C.2.

**End** The constraint in Eq. 9 is very similar to what was recently proposed in Tartaglione et al. (2021) with EnD. However, EnD lacks the additional constraint on the standard deviation of the distances, which is given by Eq. 10. An intuitive difference can be found in Fig. 3 and 4 of the Appendix. The constraints imposed by EnD (only first moments) can be fulfilled even if there is an effect of the bias features on the ordering of the positive samples. The use of constraints on the second moments, as in the proposed method, can remove the effect of the bias. An analytical comparison can be found in Appendix A.3.

**BiasCon** In Hong & Yang (2021), authors propose the BiasCon loss, which is similar to SupCon but only aligns positive bias-conflicting samples. It looks for an encoder $f$ that fulfills:

$$s_j^- - s_i^{+,b'} \leq -\epsilon \quad \forall i, j \quad \text{and} \quad s_p^{+,b} - s_i^{+,b'} \leq 0 \quad \forall i, p \text{ and } \quad s_t^{+,b'} - s_i^{+,b'} \leq 0 \quad \forall i, t \neq i \quad (13)$$

The problem here is that we try to separate the negative samples from only the positive bias-conflicting samples, ignoring the positive bias-aligned samples. This is probably why the authors proposed to combine this loss with a standard Cross Entropy.

---

[2]The same reasoning can be applied to negative samples (omitted for brevity.)

## 4 EXPERIMENTS

In this section, we describe the experiments we perform to validate our proposed losses. We perform two sets of experiments. First, we benchmark our framework, presented in Sec. 3, on standard vision datasets such as: CIFAR-10 (Krizhevsky et al., a), CIFAR-100 (Krizhevsky et al., b) and ImageNet-100 (Deng et al., 2009). Then, we analyze biased settings, employing BiasedMNIST (Bahng et al., 2020), Corrupted-CIFAR10 (Hendrycks & Dietterich, 2019), bFFHQ (Lee et al., 2021), 9-Class ImageNet (Ilyas et al., 2019) and ImageNet-A (Hendrycks et al., 2021). The code can be found at `https://github.com/EIDOSLAB/unbiased-contrastive-learning`.

### 4.1 EXPERIMENTS ON GENERIC VISION DATASETS

We conduct an empirical analysis of the $\epsilon$-SupCon and $\epsilon$-SupInfoNCE losses on standard vision datasets to evaluate the different formulations and to assess the impact of the $\epsilon$ parameter. We compare our results with baseline implementations including Cross Entropy (CE) and SupCon.

**Experimental details** We use the original setup from SupCon (Khosla et al., 2020), employing a ResNet-50, a large batch size (1024), a learning rate of 0.5, a temperature of 0.1, and multiview augmentation, for CIFAR-10 and CIFAR-100. Additional experimental details (including ImageNet-100[3]) and the different hyperparameters configurations are provided in Sec. B of the Appendix.

**Results** First, we compare our proposed $\epsilon$-SupInfoNCE loss with the $\epsilon$-SupCon loss derived in Sec. 3. As reported in Tab. 1, $\epsilon$-SupInfoNCE performs better than $\epsilon$-SupCon: we conjecture that the lack of the non-contrastive term of Eq. 5 leads to increased robustness, as it will also be shown in Sec. 4.2. For this reason, we focus on $\epsilon$-SupInfoNCE. Further comparison with different values of $\epsilon$ can be found in Sec. C.1, showing that *SupCon* $\leq \epsilon$-*SupCon* $\leq \epsilon$-*SupInfoNCE* in terms of accuracy. Results

Table 1: Comparison of $\epsilon$-SupInfoNCE and $\epsilon$-SupCon on ImageNet-100.

| Loss | Acc@1 |
|---|---|
| $\epsilon$-SupInfoNCE | **83.3**$_{\pm 0.06}$ |
| $\epsilon$-SupCon | 82.83$_{\pm 0.11}$ |

on general computer vision datasets are presented in Tab. 2, in terms of top-1 accuracy. We report the performance for the best value of $\epsilon$; the complete results can be found in Sec. C.1. The results are averaged across 3 trials for every configuration, and we also report the standard deviation. We obtain significant improvement with respect to all baselines and, most importantly, SupCon, on all benchmarks: on CIFAR-10 ($+0.5\%$), on CIFAR-100 ($+0.63\%$), and on ImageNet-100 ($+1.31\%$).

Table 2: Accuracy on vision datasets. SimCLR and Max-Margin results from Khosla et al. (2020). Results denoted with * are (re)implemented with mixed precision due to memory constraints.

| Dataset | Network | SimCLR | Max-Margin | SimCLR* | CE* | SupCon* | $\epsilon$-SupInfoNCE* |
|---|---|---|---|---|---|---|---|
| CIFAR-10 | ResNet-50 | 93.6 | 92.4 | 91.74$_{\pm 0.05}$ | 94.73$_{\pm 0.18}$ | 95.64$_{\pm 0.02}$ | **96.14**$_{\pm 0.01}$ |
| CIFAR-100 | ResNet-50 | 70.7 | 70.5 | 68.94$_{\pm 0.12}$ | 73.43$_{\pm 0.08}$ | 75.41$_{\pm 0.19}$ | **76.04**$_{\pm 0.01}$ |
| ImageNet-100 | ResNet-50 | - | - | 66.14$_{\pm 0.08}$ | 82.1$_{\pm 0.59}$ | 81.99$_{\pm 0.08}$ | **83.3**$_{\pm 0.06}$ |

### 4.2 EXPERIMENTS ON BIASED DATASETS

Next, we move on to analyzing how our proposed loss performs on biased learning settings. We employ five datasets, ranging from synthetic data to real facial images: Biased-MNIST, Corrupted-CIFAR10, bFFHQ, and 9-Class ImageNet along with ImageNet-A. The detailed setup and experimental details are provided in the appendix B.

**Biased-MNIST** is a biased version of MNIST (Deng, 2012), proposed in Bahng et al. (2020). A color bias is injected into the dataset, by colorizing the image background with ten predefined colors associated with the ten different digits. Given an image, the background is colored with the predefined color for that class with a probability $\rho$, and with any one of the other colors with a probability $1 - \rho$. Higher values of $\rho$ will lead to more biased data. In this work, we explore the datasets in

---

[3]Due to computing resources constraints, we were not able to evaluate ImageNet-1k.

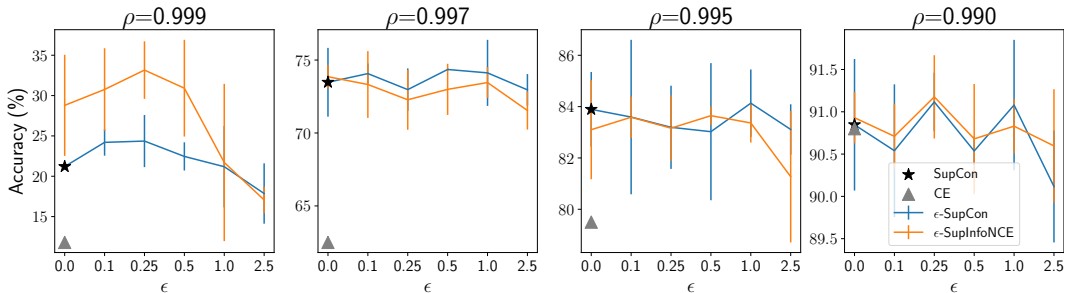

Figure 2: Comparison of $\epsilon$-SupCon and $\epsilon$-SupInfoNCE on Biased-MNIST. It is noticeable that for $\rho \leq 0.997$, $\epsilon$-SupInfoNCE and $\epsilon$-SupCon are comparable, while for $\rho = 0.999$ the gap is significantly larger: this could be due to the additional non-contrastive condition of SupCon.

different values of $\rho$: 0.999, 0.997, 0.995 and 0.99. An unbiased test set is built with $\rho = 0.1$. We compare with cross entropy baseline and with other debiasing techniques, namely EnD (Tartaglione et al., 2021), LNL (Nam et al., 2020) and BiasCon and BiasBal (Hong & Yang, 2021).

*Analysis of $\epsilon$-SupInfoNCE and $\epsilon$-SupCon:* First, we perform an evaluation of the $\epsilon$-SupCon and $\epsilon$-SupInfoNCE losses alone, without our debiasing regularization term. Fig. 2 shows the accuracy on the unbiased test set, with the different values of $\rho$. Baseline results of a cross-entropy model (CE) are reported in Tab. 3. Both losses result in higher accuracy compared to the cross entropy. The generally higher robustness of contrastive-based formulations is also confirmed by the related literature (Khosla et al., 2020). Interestingly, in the most biased setting ($\rho = 0.999$), we observe that $\epsilon$-SupInfoNCE obtains higher accuracy than $\epsilon$-SupCon. Our conjecture is that the non-contrastive term of SupCon in Eq. 5 ($s_t^+ - s_i^+ \leq 0 \quad \forall i, t$) can lead, in highly biased settings, to more biased representations as the bias-aligned samples will be especially predominant among the positives. For this reason, we focus on $\epsilon$-SupInfoNCE in the remaining of this work.

*Debiasing with FairKL:* Next, we apply our regularization technique FairKL jointly with $\epsilon$-SupInfoNCE, and compare it with the other debiasing methods. The results are shown in Tab. 3. Our technique achieves the best results in all experiments, with high gaps in accuracy, especially in the most difficult settings (lower $\rho$). For completeness, we also evaluate the debiasing power of FairKL with different losses, i.e. CE and $\epsilon$-SupCon. With FairKL we obtain better results than most of the other baselines with either CE, $\epsilon$-SupCon or $\epsilon$-SupInfoNCE; the latter achieves the best performance, confirming the results observed in Sec 4.1. For this reason, in the rest of the work, we focus on $\epsilon$-SupInfoNCE.

Table 3: Top-1 accuracy (%) on Biased-MNIST. Reference results from Hong & Yang (2021). Results denoted with * are re-implemented without color-jittering and bias-conflicting oversampling.

| Method | 0.999 | 0.997 | 0.995 | 0.99 |
|---|---|---|---|---|
| CE Hong & Yang (2021) | 11.8±0.7 | 62.5±2.9 | 79.5±0.1 | 90.8±0.3 |
| LNL Kim et al. (2019) | 18.2±1.2 | 57.2±2.2 | 72.5±0.9 | 86.0±0.2 |
| $\epsilon$-SupCon | 24.36±3.23 | 74.35±0.09 | 84.13±1.31 | 91.12±0.35 |
| $\epsilon$-SupInfoNCE | 33.16±3.57 | 73.86±0.81 | 83.65±0.36 | 91.18±0.49 |
| EnD Tartaglione et al. (2021) | 59.5±2.3 | 82.70±0.3 | 94.0±0.6 | 94.8±0.3 |
| BiasCon+BiasBal* Hong & Yang (2021) | 30.26±11.08 | 82.83±4.17 | 88.20±2.27 | 95.04±0.86 |
| BiasBal Hong & Yang (2021) | 76.8±1.6 | 91.2±0.2 | 93.9±0.1 | 96.3±0.2 |
| BiasCon+CE* Hong & Yang (2021) | 15.06±2.22 | 90.48±5.26 | 95.95±0.11 | 97.67±0.09 |
| CE + FairKL | 79.9±4.29 | 93.86±1.13 | 94.85±0.55 | 95.92±0.17 |
| $\epsilon$-SupCon + FairKL | 89.45±1.82 | 95.75±0.16 | 96.31±0.81 | 96.72±0.2 |
| $\epsilon$-SupInfoNCE + FairKL | **90.51**±1.55 | **96.19**±0.23 | **97.00**±0.06 | **97.86**±0.02 |

**Corrupted CIFAR-10** is built from the CIFAR-10 dataset, by correlating each class with a certain texture (brightness, frost, etc.) following the protocol proposed in Hendrycks & Dietterich (2019). Similarly to Biased-MNIST, the dataset is provided with five different levels of ratio between bias-

conflicting and bias-aligned samples. The results are shown in Tab. 4. Notably, we obtain the best results in the most difficult scenario, when the amount of bias-conflicting samples is the lowest. Again, for the other settings, we obtain comparable results with the state of the art.

**bFFHQ** is proposed by Lee et al. (2021), and contains facial images. They construct the dataset in such a way that most of the females are young (age range 10-29), while most of the males are older (age range 40-59). The ratio between bias-conflicting and bias-aligned provided for this dataset is 0.5. The results are shown in Tab. 4, where our technique outperforms all other methods.

**9-Class ImageNet and ImageNet-A** We also test our method on the more complex and realistic 9-Class ImageNet (Ilyas et al., 2019) dataset. This dataset is a subset of ImageNet, which is known to contain textural biases (Geirhos et al., 2019). It aggregates 42 of the original classes into 9 macro categories. Following Hong & Yang (2021), we train a BagNet18 (Brendel & Bethge, 2019) as the bias-capturing model, which we then use to compute a bias score for the training samples, to apply within our regularization term. More details and the experimental setup can be found in the Sec. B.2.4. We evaluate the accuracy on the test set (biased) along with the unbiased accuracy (UNB), computed with the texture labels assigned in Brendel & Bethge (2019). We also report accuracy results on ImageNet-A (IN-A) dataset, which contains bias-conflicting samples (Hendrycks et al., 2021). Results are shown in Tab. 5. On the biased test set, the results are comparable with SoftCon, while on the harder sets unbiased and ImageNet-A we achieve SOTA results.

Table 4: Top-1 accuracy (%) on Corrupted CIFAR-10 with different corruption ratio (%) and on bFFHQ. Reference results are taken from Lee et al. (2021).

| | Corrupted CIFAR-10 | | | | bFFHQ |
| | Ratio | | | | Ratio |
| Method | 0.5 | 1.0 | 2.0 | 5.0 | 0.5 |
|---|---|---|---|---|---|
| Vanilla Lee et al. (2021) | $23.08_{\pm 1.25}$ | $25.82_{\pm 0.33}$ | $30.06_{\pm 0.71}$ | $39.42_{\pm 0.64}$ | $56.87_{\pm 2.69}$ |
| EnD Tartaglione et al. (2021) | $19.38_{\pm 1.36}$ | $23.12_{\pm 1.07}$ | $34.07_{\pm 4.81}$ | $36.57_{\pm 3.98}$ | $56.87_{\pm 1.42}$ |
| HEX Wang et al. (2019a) | $13.87_{\pm 0.06}$ | $14.81_{\pm 0.42}$ | $15.20_{\pm 0.54}$ | $16.04_{\pm 0.63}$ | $52.83_{\pm 0.90}$ |
| ReBias Bahng et al. (2020) | $22.27_{\pm 0.41}$ | $25.72_{\pm 0.20}$ | $31.66_{\pm 0.43}$ | $43.43_{\pm 0.41}$ | $59.46_{\pm 0.64}$ |
| LfF Nam et al. (2020) | $28.57_{\pm 1.30}$ | $33.07_{\pm 0.77}$ | $39.91_{\pm 0.30}$ | $50.27_{\pm 1.56}$ | $62.2_{\pm 1.0}$ |
| DFA Lee et al. (2021) | $29.95_{\pm 0.71}$ | $36.49_{\pm 1.79}$ | $\mathbf{41.78}_{\pm 2.29}$ | $\mathbf{51.13}_{\pm 1.28}$ | $\underline{63.87}_{\pm 0.31}$ |
| $\epsilon$-SupInfoNCE + FairKL | $\mathbf{33.33}_{\pm 0.38}$ | $\mathbf{36.53}_{\pm 0.38}$ | $\underline{41.45}_{\pm 0.42}$ | $\underline{50.73}_{\pm 0.90}$ | $\mathbf{64.8}_{\pm 0.43}$ |

Table 5: Top-1 accuracy (%) on 9-Class ImageNet biased and unbiased (UNB) sets, and ImageNet-A (IN-A). Reference results from Hong & Yang (2021).

| | Vanilla | SIN | LM | RUBi | ReBias | LfF | SoftCon | $\epsilon$-SupInfoNCE + FairKL |
|---|---|---|---|---|---|---|---|---|
| Biased | $94.0_{\pm 0.1}$ | $88.4_{\pm 0.9}$ | $79.2_{\pm 1.1}$ | $93.9_{\pm 0.2}$ | $94.0_{\pm 0.2}$ | $91.2_{\pm 0.1}$ | $\mathbf{95.3}_{\pm 0.2}$ | $\underline{95.1}_{\pm 0.1}$ |
| UNB | $92.7_{\pm 0.2}$ | $86.6_{\pm 1.0}$ | $76.6_{\pm 1.2}$ | $92.5_{\pm 0.2}$ | $92.7_{\pm 0.2}$ | $89.6_{\pm 0.3}$ | $\underline{94.1}_{\pm 0.3}$ | $\mathbf{94.8}_{\pm 0.3}$ |
| IN-A | $30.5_{\pm 0.5}$ | $24.6_{\pm 2.4}$ | $19.0_{\pm 1.2}$ | $31.0_{\pm 0.2}$ | $30.5_{\pm 0.2}$ | $29.4_{\pm 0.8}$ | $\underline{34.1}_{\pm 0.6}$ | $\mathbf{35.7}_{\pm 0.5}$ |

## 5 CONCLUSIONS

In this work, we propose a metric-learning-based framework for supervised representation learning. We propose a new loss, called *$\epsilon$-SupInfoNCE*, that is based on the definition of the $\epsilon$-margin, which is the minimal margin between positive and negative samples. By adjusting this value, we are able to find a tighter approximation of the mutual information and achieve better results compared to standard Cross-Entropy and to the SupCon loss. Then, we tackle the problem of learning unbiased representations when the training data contains strong biases. This represents a failure case for InfoNCE-like losses. We propose *FairKL*, a debiasing regularization term derived from our framework. With it, we enforce equality between the distribution of distances of bias-conflicting samples and bias-aligned samples. This, together with the increase of the $\epsilon$ margin, allows us to reach state-of-the-art performances in the most extreme cases of biases in different datasets, comprising both synthetic data and real-world images.

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

## A  THEORETICAL RESULTS

### A.1  COMPLETE DERIVATIONS FOR SECTION 3

In this section, we present the complete analytical derivation for the equations found in Sec. 3. All of the presented derivations are based on the smooth max approximation with the LogSumExp (LSE) operator:

$$\max(x_1, x_2, ..., x_3) \approx \log(\sum_i \exp(x_i)) \tag{14}$$

#### A.1.1  COMPUTATIONS OF $\epsilon$-INFONCE (2)

We consider Eq. 2 and we obtain:

$$\arg\min_f \max(-\epsilon, \{s_j^- - s^+\}_{j=1,...,N}) \approx \arg\min_f \underbrace{\left[ -\log\left( \frac{\exp(s^+)}{\exp(s^+ - \epsilon) + \sum_j \exp(s_j^-)} \right) \right]}_{\epsilon - InfoNCE} \tag{15}$$

Starting from the left-hand side, we have:

$$
\begin{aligned}
\max(-\epsilon, \{s_j^- - s^+\}_{j=1,...,N}) &\approx \log\left( \exp(-\epsilon) + \sum_j \exp(s_j^- - s^+) \right) \\
&= \log\left( \exp(-\epsilon) + \exp(-s^+) \sum_j \exp(s_j^-) \right) \\
&= \log\left( \exp(-s^+)\left( \exp(s^+ - \epsilon) + \sum_j \exp(s_j^-) \right) \right) \\
&= \log\exp(-s^+) + \log\left( \exp(s^+ - \epsilon) + \sum_j \exp(s_j^-) \right) \\
&= \underbrace{-\log\left( \frac{\exp(s^+)}{\exp(s^+ - \epsilon) + \sum_j \exp(s_j^-)} \right)}_{\epsilon - InfoNCE}
\end{aligned}
\tag{16}
$$

#### A.1.2  MULTIPLE POSITIVE EXTENSION

Extending Eq. 2 to multiple positives can be done in different ways. Here, we list four possible choices. Empirically, we found that solution c) gave the best results and is the most convenient to implement for efficiency reasons.

$$a) \max(-\epsilon, \{s_j^- - s_i^+\}_{\substack{i=1,\ldots,P \\ j=1,\ldots,N}}) = -\log\left(\frac{\exp(\sum_i s_i^+)}{\exp(\sum_i s_i^+ - \epsilon) + (\sum_j \exp(s_j^-))(\sum_i \exp(\sum_{t\neq i} s_t^+))}\right)$$

$$b) \sum_j \max(-\epsilon, \{s_j^- - s_i^+\}_{i=1,\ldots,P}) = -\sum_j \log\left(\frac{\exp(\sum_i s_i^+)}{\exp(\sum_i s_i^+ - \epsilon) + \exp(s_j^-)(\sum_i \exp(\sum_{t\neq i} s_t^+))}\right)$$

$$c) \sum_i \max(-\epsilon, \{s_j^- - s_i^+\}_{j=1,\ldots,N}) = -\sum_i \log\left(\frac{\exp(s^+)}{\exp(s^+ - \epsilon) + \sum_j \exp(s_j^-)}\right)$$

$$d) \sum_i \sum_j \max(-\epsilon, s_j^- - s_i^+) = -\sum_i \sum_j \log\left(\frac{\exp(s^+)}{\exp(s^+ - \epsilon) + \exp(s_j^-)}\right)$$

$$(17)$$

### A.1.3 COMPUTATIONS OF $\epsilon$-SUPINFONCE (4)

The computations are very similar to Eq. 16. We obtain:

$$\arg\min_f \sum_i \max(-\epsilon, \{s_j^- - s_i^+\}_{j=1,\ldots,N}) \approx \arg\min_f \left[\sum_i \log\left(\exp(-\epsilon) + \sum_j \exp(s_j^- - s_i^+)\right)\right]$$

$$(18)$$

Starting from the left-hand side, we have:

$$\sum_i \max(-\epsilon, \{s_j^- - s_i^+\}_{j=1,\ldots,N}) \approx \sum_i \log\left(\exp(-\epsilon) + \sum_j \exp(s_j^- - s_i^+)\right)$$

$$= \sum_i \log\left(\exp(-\epsilon) + \frac{\sum_j \exp(s_j^-)}{\exp(s_i^+)}\right)$$

$$= \sum_i \log\left(\frac{\exp(s_i^+ - \epsilon)\sum_j \exp(s_j^-)}{\exp(s_i^+)}\right) \qquad (19)$$

$$= \underbrace{-\sum_i \log\left(\frac{\exp(s_i^+)}{\exp(s_i^+ - \epsilon)\sum_j \exp(s_j^-)}\right)}_{\epsilon-SupInfoNCE}$$

### A.1.4 COMPUTATIONS OF $\epsilon$-SUPCON (5)

We extend Eq. 4 by adding the non contrastive conditions:

$$s_j^- - s_i^+ \leq -\epsilon \quad \forall i,j \quad \text{and} \quad s_t^+ - s_i^+ \leq 0 \quad \forall i, t \neq i \qquad (20)$$

and we show

$$\frac{1}{P}\sum_i \max(0, \{s_j^- - s_i^+ + \epsilon\}_j, \{s_t^+ - s_i^+\}_{t\neq i}) \approx \epsilon - \underbrace{\frac{1}{P}\sum_i \log\left(\frac{\exp(s_i^+)}{\sum_t \exp(s_i^+ - \epsilon) + \sum_j \exp(s_j^-)}\right)}_{\epsilon-SupCon}$$

$$(21)$$

Starting from the left-hand side, we have:

$$\frac{1}{P}\sum_i \max(0, \{s_j^- - s_i^+ + \epsilon\}_{j=1,\dots,N}, \{s_t^+ - s_i^+\}_{t \neq i}) \approx \frac{1}{P}\sum_i \log\left(1 + \sum_j \exp(s_j^- - s_i^+ + \epsilon) + \sum_{t \neq i}\exp(s_t^+ - s_i^+)\right)$$

$$= = \frac{1}{P}\sum_i \log\left(1 + \frac{\sum_j \exp(s_j^-)}{\exp(s_i^+ - \epsilon)} + \frac{\sum_{t\neq i}\exp(s_t^+)}{\exp(s_i^+)}\right)$$

$$= \frac{1}{P}\sum_i \log\left(\frac{\exp(s_i^+ - \epsilon) + \sum_j \exp(s_j^-) + \sum_{t\neq i}\exp(s_t^+ - \epsilon)}{\exp(s_i^+ - \epsilon)}\right)$$

$$= -\frac{1}{P}\sum_i \log\left(\frac{\exp(s_i^+ - \epsilon)}{\sum_t \exp(s_t^+ - \epsilon) + \sum_j \exp(s_j^-)}\right)$$

$$= \epsilon - \underbrace{\frac{1}{P}\sum_i \log\left(\frac{\exp(s_i^+)}{\sum_t \exp(s_t^+ - \epsilon) + \sum_j \exp(s_j^-)}\right)}_{\epsilon-SupCon}$$

$$(22)$$

### A.1.5 COMPUTATIONS OF $\mathcal{L}_{in}^{sup}$ (7)

Here we show that:

$$\max(s_j^-) < \max(s_i^+) \approx \underbrace{-\log\left(\sum_i \frac{\exp(s_i^+)}{\sum_t \exp(s_t^+) + \sum_j \exp(s_j^-)}\right)}_{\mathcal{L}_{in}^{sup}} \tag{23}$$

Starting from the left-hand side, and given that:

$$\max(s_j^-) < \max(s_i^+) \approx \log(\sum_j \exp(s_j^-)) - \log(\sum_i \exp(s_i^+)) < 0$$

we have:

$$\max\left(0, \log(\sum_j \exp(s_j^-)) - \log(\sum_i \exp(s_i^+))\right) \approx \log\left(1 + \exp\left(\log(\sum_j \exp(s_j^-)) - \log(\sum_i \exp(s_i^+))\right)\right)$$

$$= \log\left(1 + \exp\left(\log\left(\frac{\sum_j \exp(s_j^-)}{\sum_i \exp(s_i^+)}\right)\right)\right)$$

$$= \log\left(1 + \frac{\sum_j \exp(s_j^-))}{\sum_i \exp(s_i^+)}\right)$$

$$= \log\left(\frac{\sum_i \exp(s_i^+) + \sum_j \exp(s_j^-))}{\sum_i \exp(s_i^+)}\right)$$

$$= -\log\underbrace{\left(\sum_i \frac{\exp(s_i^+)}{\sum_t \exp(s_t^+) + \sum_j \exp(s_j^-))}\right)}_{\mathcal{L}_{in}^{sup}}$$

$$(24)$$

### A.1.6 COMPUTATIONS OF EQ.17-A

$$\arg\min_f \max(-\epsilon, \{s_j^- - s_i^+\}_{\substack{i=1,...,P \\ j=1,...,N}}) \approx \arg\min_f \log\left(\exp(-\epsilon) + \sum_i \sum_j \exp(s_j^- - s_i^+)\right) \quad (25)$$

We have:

$$\mathcal{L} = \log\left(\exp(-\epsilon) + \sum_i^P \left(\frac{\sum_j \exp(s_j^-)}{\exp(s_i^+))}\right)\right) = \log\left(\sum_i^P \left(\frac{\exp(-\epsilon)}{P} + \frac{\sum_j \exp(s_j^-)}{\exp(s_i^+))}\right)\right)$$

$$= \log\left(\sum_i^P \left(\frac{\exp(s_i^+ - \epsilon) + P(\sum_j \exp(s_j^-))}{P(\exp(s_i^+)))}\right)\right) = \log\left(\sum_i \left(\frac{\frac{1}{P}\exp(s_i^+ - \epsilon)) + \sum_j \exp(s_j^-)}{\exp(s_i^+)}\right)\right)$$

$$= \log\left(\frac{\sum_i \left[\frac{1}{P}\exp(s_i^+ - \epsilon)(\prod_{t \neq i}\exp(s_t^+)) + (\sum_j \exp(s_j^-))(\prod_{t \neq i}\exp(s_t^+))\right]}{\prod_i \exp(s_i^+)}\right)$$

$$= \log\left(\frac{\exp(-\epsilon)\prod_i \exp(s_i^+) + (\sum_j \exp(s_j^-))(\sum_i \prod_{t \neq i}\exp(s_t^+))}{\prod_i \exp(s_i^+)}\right)$$

$$= -\log\left(\frac{\exp(\sum_i s_i^+)}{\exp(\sum_i s_i^+ - \epsilon) + (\sum_j \exp(s_j^-))(\sum_i \exp(\sum_{t \neq i} s_t^+))}\right)$$

$$(26)$$

### A.2 BOUNDNESS OF THE $\epsilon$-MARGIN

In this section, we give insights on how an optimal value of $\epsilon$ can be estimated. First of all, it is easy to show that $\epsilon$ is bounded and cannot grow to infinity. Given that $||f(x)||_2 = 1$, then we

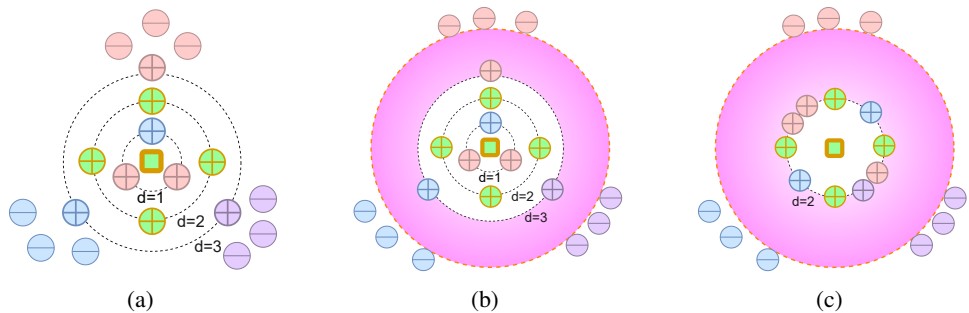

(a)  (b)  (c)

Figure 3: When considering only Eq. 9 a-left (average of distances) or Eq. 9a-right (average of similarities), that is when using EnD Tartaglione et al. (2021), we may obtain a suboptimal configuration such as (a), where we can still (partially) order the distances of positive samples from the anchor based on the bias features. We can see that the conditions in Eq. 9 are fulfilled, namely the average of the distances of bias-aligned and bias-conflicting samples from the anchor are the same ($\mu_{+,b} = \mu_{+,b'} = 2$). This is only partially mitigated when using a margin $\epsilon > 0$ (b). However, the standard deviations of the distances of bias-aligned and bias-conflicting samples in (a) and (b) are different ($\sigma_{+,b} = 0$, while $\sigma_{+,b'} = 1$). This can be computed using the distances $d$ reported in the figure. If we also consider the conditions on the standard deviations of the distances, as proposed in FairKL (Eq. 10), the ordering is removed and thus also the effect of the bias (c). In (c), we show the case in which both mean and standard deviation of the distributions match (in a simplified case with $\sigma$=0). A simulated example is shown in Fig. 4.

$\max(d^+ - d^-) = 2$. This is the case in which the two samples are aligned at opposite poles of the hypersphere. We can conclude that, in general, $\epsilon$ will be less than 2. If we also take into account the temperature $\tau$, when $\epsilon \leq 2/\tau$. This is always true, however, a stricter upper bound can be found if we consider the geometric properties of the latent space. For example, Graf et al. (2021) show that when the SupCon loss converges to its minimum value, then the representations of the different classes are aligned on a regular simplex. This property could be used to compute a precise upper bound of the $\epsilon$ margin, depending on the number of classes in the dataset. We leave further analysis on this matter as future work.

A.3  THEORETICAL COMPARISON WITH ENED

Here, we present a more detailed theoretical analysis of EnD (Tartaglione et al., 2021), and we show that the EnD regularization term can be equivalent to the condition in Eq. 9:

$$a)\frac{1}{P_c}\sum_k |s_k^{+,b'}| - \frac{1}{P_a}\sum_i |s_i^{+,b}| = 0 \quad b)\frac{1}{N_c}\sum_t |s_t^{-,b'}| - \frac{1}{N_a}\sum_j |s_j^{-,b}| = 0 \qquad (27)$$

which can be turned into a minimization term $\mathcal{R}$, using the method of Lagrange multipliers:

$$\mathcal{R} = -\lambda_1 \left( \frac{1}{P_c}\sum_k |s_k^{+,b'}| - \frac{1}{P_a}\sum_i |s_i^{+,b}| \right) - \lambda_2 \left( \frac{1}{N_c}\sum_t |s_t^{-,b'}| - \frac{1}{N_a}\sum_j |s_j^{-,b}| \right) \qquad (28)$$

Now, if we assume $\lambda_1 = \lambda_2 = 1$, we can re-arrange the terms, obtaining:

$$\mathcal{R} = \underbrace{\left( \frac{1}{P_a}\sum_i |s_i^{+,b}| + \frac{1}{N_a}\sum_j |s_j^{-,b}| \right)}_{\mathcal{R}^\perp} - \underbrace{\left( \frac{1}{P_c}\sum_k |s_k^{+,b'}| + \frac{1}{N_c}\sum_t |s_t^{-,b'}| \right)}_{\mathcal{R}^\parallel} \qquad (29)$$

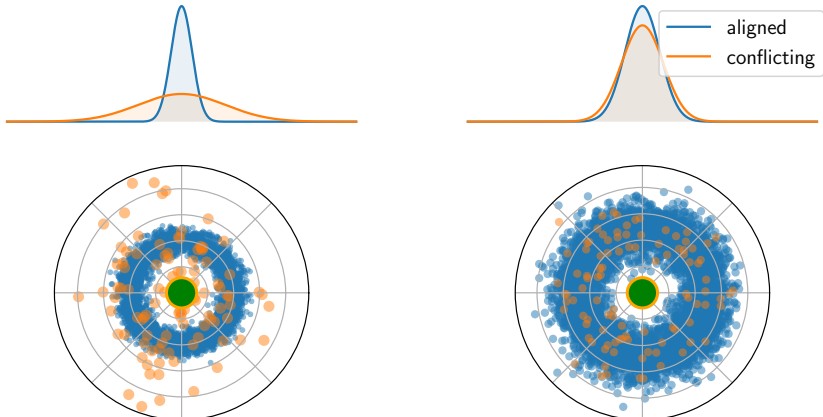

Figure 4: Toy example with simulated data to better explain the suboptimal solution of Fig. 3. We make the hypothesis that the distributions of the distances do follow a Gaussian distribution. In blue and in orange are shown the bias-aligned and the bias-conflicting samples respectively. The green sample represents the anchor. On the left, data points are sampled from two normal distributions with the same mean but *different* std. We can see that the two distributions do not match. This shows that, even if the first order constraints of EnD are fulfilled, there might still be an effect of the bias. On the contrary, on the right, the two distributions have almost the same statistics (both average and std) and the KL divergence is almost 0. In that case, the bias effect is basically removed.

The two terms we obtain are equivalent to the disentangling term $\mathcal{R}^\perp$ and to the entangling term $\mathcal{R}^\parallel$ of EnD: $\mathcal{R}^\perp$ tries to decorrelate all of the samples which share the same bias attribute, while the $\mathcal{R}^\parallel$ tries to maximize the correlation of samples which belong to the same class but have different bias attributes. However, some practical differences between the two formulation remain in how the different terms are weighted: for Eq. 9 we would have:

$$\mathcal{R} = \lambda_1(\mu_{+,b'} - \mu_{+,b}) + \lambda_2(\mu_{-,b'} - \mu_{-,b}) \tag{30}$$

while for EnD we would have:

$$\mathcal{R} = \lambda_1(\mu_{+,b} - \mu_{-,b}) + \lambda_2(\mu_{+,b'} - \mu_{-,b'}) \tag{31}$$

## B    EXPERIMENTAL SETUP

All of our experiments were run using PyTorch 1.10.0. We used a cluster with 4 NVIDIA V100 GPUs and a cluster of 8 NVIDIA A40 GPUs. For consistency, when training with constrastive losses we use a temperature value $\tau = 0.1$ across all of our experiments.

### B.1    GENERIC VISION DATASETS

#### B.1.1    CIFAR-10 AND CIFAR-100

We use the original setup from SupCon (Khosla et al., 2020), employing a ResNet-50, large batch size (1024), learning rate of 0.5, temperature of 0.1 and multiview augmentation, for CIFAR-10 and CIFAR-100. We use SGD as optimizer with a momentum of 0.9, and train for 1000 epochs. Learning rate is decayed with a cosine policy with warmup from 0.01, with 10 warmup epochs.

#### B.1.2    IMAGENET-100

For ImageNet-100 we employ the ResNet50 architectures (He et al., 2015). We use SGD as optimizer with a weight decay of $10^{-4}$ and momentum of 0.9, with an initial learning rate of 0.1. We

train for 100 epochs with a batch size of 512, and we decay the learning rate by a factor of 0.1 every 30 epochs.

## B.2 BIASED DATASETS

When employing our debiasing term, we find that scaling the $\epsilon$-SupInfoNCE loss by a small factor $\alpha$ ($\leq 1$) and using $\lambda$ closer to 1, is stabler then using values of $\lambda >> 1$ (as done in Tartaglione et al. (2021)) and tends to produce better results. For biased datasets, we do not make use of the projection head used in Khosla et al. (2020); Chen et al. (2020). For this reason, we also avoid the aggressive augmentation usually employed by contrastive methods (more on this in Sec. C.3). Furthermore, as also done by Hong & Yang (2021), we also experimented with a small contribution of the cross entropy loss for training the model end-to-end; however, we did not find any benefit in doing so, compared to training a linear classifier separately.

### B.2.1 BIASED-MNIST

We emply the network architecture *SimpleConvNet* proposed by Bahng et al. (2020), consisting of four convolutional layers with $7 \times 7$ kernels. We use the Adam optimizer with a learning rate of 0.001, a weight decay of $10^{-5}$ and a batch size of 256. We decay the learning rate by a factor of 0.1 at $1/3$ and $2/3$ of the epochs (26 and 53). We train for 80 epochs.

**Hyperparameters configuration** The hyperparameters for Tab. 3 are reported in Tab. 6.

Table 6: Biased-MNIST hyperparameters of Tab. 3

|  |  | Corr ($\rho$) | | | |
|  |  | 0.999 | 0.997 | 0.995 | 0.990 |
| --- | --- | --- | --- | --- | --- |
| $\epsilon$-SupCon | $\alpha$ | 0.03 | 0.03 | 0.03 | 0.03 |
|  | $\lambda$ | 0.75 | 0.5 | 0.5 | 0.5 |
|  | $\epsilon$ | 0.25 | 0 | 0.5 | 0 |
| $\epsilon$-SupInfoNCE | $\alpha$ | 0.03 | 0.03 | 0.03 | 0.03 |
|  | $\lambda$ | 0.75 | 0.75 | 0.75 | 0.5 |
|  | $\epsilon$ | 0.5 | 0.5 | 0.5 | 0.5 |

### B.2.2 CORRUPTED CIFAR-10

For this dataset we employ the ResNet-18 architecture. We use the Adam optimizer, with an initial learning rate of 0.001, a weight decay of 0.0001. The other Adam parameters are the pytorch default ones ($\beta_1 = 0.9$, $\beta_2 = 0.999$, $\epsilon = 10^{-8}$). We train for 200 epochs with a batch size of 256. We decay the learning rate using a cosine annealing policy.

**Hyperparameters configuration:** Table 7 shows the hyperparameters for the results reported in Tab. 4 of the main paper.

Table 7: Corrupted CIFAR-10 hyperparameters

|  | Ratio (%) | | | |
|  | 0.5 | 1.0 | 2.0 | 5.0 |
| --- | --- | --- | --- | --- |
| $\alpha$ | 0.1 | 0.1 | 0.1 | 0.1 |
| $\lambda$ | 1.0 | 1.0 | 1.0 | 1.0 |
| $\epsilon$ | 0.1 | 0.25 | 0.5 | 0.25 |

### B.2.3 BFFHQ

Following Lee et al. (2021),we use the ResNet-18 architecture. We use the Adam optimizer, with an initial learning rate of 0.0001, and train for 100 epochs. For this experiment, we set $\alpha = 0.1$,

$\epsilon = 0.25$ and $\lambda = 1.5$. Differently from Lee et al. (2021) we use a batch size of 256 (vs 64) as contrastive losses benefit more from larger batch sizes (Chen et al., 2020; Khosla et al., 2020). Additionally, we also use a weight decay of $10^{-4}$, rather than 0. These changes do not provide advantages to the debiasing task: we obtain an accuracy of 54.8% without FairKL, which is in line with the 56.87% reported for the vanilla model.

### B.2.4  9-CLASS IMAGENET AND IMAGENET-A

Our proposed method can also be applied in cases in which no prior label about the bias attributes is given, with only a slight change in formulation. For example, for ImageNet. Similarly to other works (Nam et al., 2020; Hong & Yang, 2021) we exploit a bias-capturing model for this purpose.

**FairKL with bias-capturing model** To use a continuous score, rather than a discrete bias label, we compute the similarity of the bias features $\tilde{b}_i^+ = s(g(x), g(x_i^+))$, where $g(\cdot)$ is the bias-capturing BagNet18 model. The bias similarity $\tilde{b}_i$ is used to obtain a weighted sample similarity: $\tilde{s}_i^{+,b} = s_i^+ \tilde{b}_i^+$ for bias-aligned samples, and $\hat{s}_i^{+,b'} = s_i^+ (1 - \tilde{b}_i^+)$ for bias-conflicting. By doing so, for example, the terms $\mu_{+,b} = \frac{1}{P_a} \sum_i d_i^{+,b}$ and $\mu_{+,b'} = \frac{1}{P_c} \sum_k d_k^{+,b'}$ become $\hat{\mu}_{+,b} = \frac{1}{N} \sum_i d_i^+ \hat{b}_i^+$ and $\hat{\mu}_{+,b'} = \frac{1}{N} \sum_i d_i^+ (1 - \hat{b}_i^+)$, where N is the batch size.

**Setup** We pretrain the bias-capturing model BagNet18 (Brendel & Bethge, 2019) for 120 epochs. For the main model ResNet18, we use the Adam optimizer, with learning rate 0.001, $\beta_1 = 0.9$ and $\beta_2 = 0.999$, weight decay of 0.0001 and a cosine decay of the learning rate. We use a batch size of 256 and train for 200 epochs. We employ as augmentation: random resized crop, random flip, and, as done in Hong & Yang (2021) random color jitter and random gray scale ($p = 0.2$). We use $\epsilon = 0.5$ and $\lambda = 1$. Given the higher complexity of this dataset, we employ $\alpha = 0.5$.

## C  ADDITIONAL EXPERIMENTS

In this section, we present some additional experiments we conducted, for a more in depth analysis of our proposed framework.

### C.1  COMPLETE RESULTS FOR COMMON VISION DATASETS

In Table 8 we report the results on CIFAR-10, CIFAR-100 and ImageNet-100 for different values of $\epsilon$. In Table 9 the full comparison between $\epsilon$-SupCon and $\epsilon$-SupInfoNCE on ImageNet-100 is presented. Our proposed $\epsilon$-SupInfoNCE outperforms SupCon in all datasets for all the $\epsilon$ values, reaching the best results. Furthermore, on ImageNet-100, we observe that the lowest accuracy obtained by $\epsilon$-SupInfoNCE (83.02%) is still higher than the best accuracy obtained by $\epsilon$-SupCon (82.83%) on the same dataset, even though $\epsilon$-SupCon is always higher than SupCon. In terms of accuracy, the results in Tab. 9 show that $SupCon \leq \epsilon - SupCon \leq \epsilon - SupInfoNCE$.

Table 8: Complete results for common vision datasets, for different values of $\epsilon$, in terms of top-1 accuracy (%). With every value of $\epsilon$ we obtain better results than SupCon (and CE) on the same dataset.

| Dataset | CE | SupCon | $\epsilon$-SupInfoNCE | | |
|---|---|---|---|---|---|
| | | | $\epsilon = 0.1$ | $\epsilon = 0.25$ | $\epsilon = 0.5$ |
| CIFAR-10 | $94.73_{\pm0.18}$ | $95.64_{\pm0.02}$ | $95.93_{\pm0.02}$ | $\mathbf{96.14}_{\pm0.01}$ | $\underline{95.95}_{\pm0.12}$ |
| CIFAR-100 | $73.43_{\pm0.08}$ | $75.41_{\pm0.19}$ | $75.85_{\pm0.07}$ | $\mathbf{76.04}_{\pm0.01}$ | $\underline{75.99}_{\pm0.06}$ |
| ImageNet-100 | $82.1_{\pm0.59}$ | $81.99_{\pm0.08}$ | $\underline{83.25}_{\pm0.39}$ | $83.02_{\pm0.41}$ | $\mathbf{83.3}_{\pm0.06}$ |

### C.2  ANALYSIS OF $\epsilon$-SUPCON FOR DEBIASING

We perform a more in-depth analysis of the debiasing capabilities of $\epsilon$-SupInfoNCE and $\epsilon$-SupCon. In Sec. 4.2 of the main text, we hypothesize that the non-constrastive condition of Eq. 20

$$s_i^+ - s_j^+ \leq 0 \quad \forall i, t \neq i$$

Table 9: Complete comparison of $\epsilon$-SupInfoNCE and $\epsilon$-SupCon on ImageNet-100 in terms of top-1 accuracy (%). The results of $\epsilon$-SupInfoNCE are higher than any results of $\epsilon$-SupCon.

| Loss | $\epsilon = 0.1$ | $\epsilon = 0.25$ | $\epsilon = 0.5$ |
|---|---|---|---|
| $\epsilon$-SupInfoNCE | **83.25**±0.39 | **83.02**±0.41 | **83.3**±0.06 |
| $\epsilon$-SupCon | 82.83±0.11 | 82.54±0.09 | 82.77±0.14 |

might actually be the reason for the loss of accuracy in $\epsilon$-SupCon when compared to $\epsilon$-InfoNCE, as shown on the analysis on Biased-MNIST in Fig. 2 of the main text.

In this section, we provide more empirical insights supporting this hypothesis. We plot the similarity of bias-aligned samples ($s^{+,b}$) and bias-conflicting samples ($s^{+,b'}$) during training, to understand how they are affected. Fig. 5 shows the bivariate histogram of the similarities obtained with the two loss functions, at different training epochs and values of $\epsilon$, on Biased-MNIST, with a training $\rho$ of 0.999. Focusing on the bias-aligned samples (first two columns), we observe that, in both cases, most values are close to 1. However, while this is true for most of the shown histograms, the presence of the non-contrastive condition of Eq. 20 produces a much tighter distribution for $\epsilon$-SupCon, when compared to $\epsilon$-SupInfoNCE. In fact, with $\epsilon$-SupInfoNCE we obtain significantly more bias-aligned samples with a similarity smaller than 1. This is especially evident if we focus on the last training epochs.

More interestingly, if we focus on the bias-conflicting similarities (last two columns), we can also notice how, on average, the distribution of similarities of bias-conflicting samples for $\epsilon$-SupCon tends to be more concentrated around the value of 0. This means that bias-conflicting samples have dissimilar representations even if they are both positives and should be mapped to the same point in the representation space. The effect of the bias is thus still quite important and it has not been discarded. On the other hand, with $\epsilon$-SupInfoNCE, we obtain a much more spread distribution, especially as the number of training epochs increases. This means that a higher number of bias-conflicting samples have a greater similarity (in the representation space), leading to more robust representations.

Clearly, $\epsilon$-SupCon focuses more on bias-aligned samples as most of them have a similarity close to 1, whereas most of the bias-conflicting samples have a similarity close to 0. With our proposed loss $\epsilon$-SupInfoNCE, this behavior is less pronounced, as the lack of the non-contrastive condition leads the model to be less focused on bias-aligned samples. This could explain why $\epsilon$-SupInfoNCE can perform better than $\epsilon$-SupCon in highly biased settings.

### C.3 TRAINING WITH A PROJECTION HEAD

In Tab. 10 we show the results on Corrupted CIFAR-10 with and without using a projection head. When employing a projection head, the loss term and the regularization are applied on the projected and original space respectively, and the final classification is performed in the original latent space before the projection head. We conjecture that the loss in accuracy is likely due i.) to the absence of the aggressive augmentation typically used for generating multiviews in contrastive setups, which are probably attenuated by the projection head ii.) minimizing $\epsilon$-SupInfoNCE and the FairKL term on the same latent space rather than two different ones, could be more beneficial for the optimization process.

Table 10: Accuracy on Corrupted CIFAR-10 with and without projection head

| | Ratio (%) | | | |
|---|---|---|---|---|
| | 0.5 | 1.0 | 2.0 | 5.0 |
| With Head | 30.85±0.19 | 32.75±0.57 | 37.95±0.14 | 45.67±0.66 |
| Without Head | **33.33**±0.38 | **36.53**±0.38 | **41.45**±0.42 | **50.73**±0.90 |

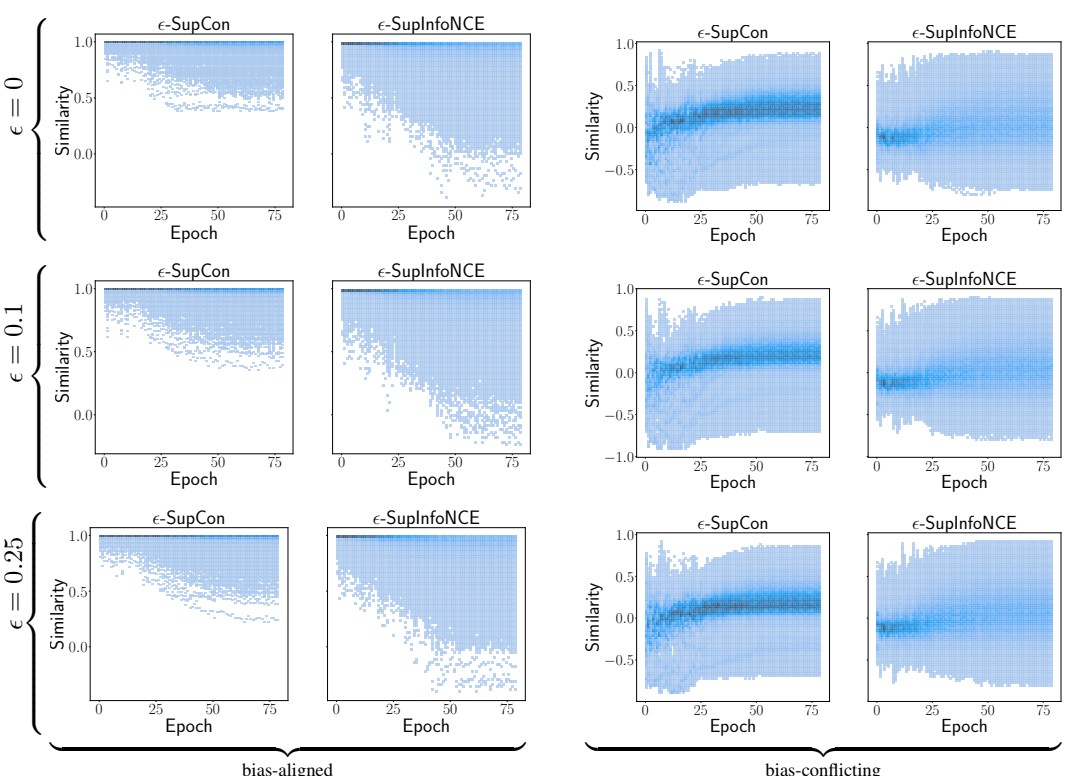

Figure 5: (*first and second columns*) Distribution of positive bias-aligned similarities $s^{+,b}$. Here $\epsilon$-SupCon tends to produce a much tighter distribution, with similarities close to 1; (*third and fourth columns*) Distribution of positive bias-conflicting similarities $s^{+,b'}$. Here $\epsilon$-SupInfoNCE, even if marginally, is able to increase the number of similar bias-conflicting samples. $\epsilon$-SupCon focuses more on bias-aligned samples, resulting in more biased representations. With $\epsilon$-SupInfoNCE, this behavior is less pronounced, as the lack of the non-contrastive condition leads to be less focused on bias-aligned samples and more focused on the bias-conflicting ones. We hypothesize that this is the reason $\epsilon$-SupInfoNCE appears to obtain better results than $\epsilon$-SupCon in more biased datasets.

## C.4 ABLATION STUDY OF DEBIASING REGULARIZATION

We perform an ablation study of our debiasing regularization on Corrupted CIFAR-10 and on bFFHQ. We test two variants of the regularization term:

1. Only with the conditions on the mean of the representations $\mu_+$ and $\mu_-$ (Eq. equation 9-a and equation 9-b), similarly to (Tartaglione et al., 2021), but with the differences in formulations of Sec. A.3;

2. Full FairKL debiasing term of Eq. 9-c and Eq. 9-d.

The results are shown in Tab. 11. As it can be easily observed, employing the full regularization constraint consistently results in better accuracy.

Table 11: Ablation study of $\mathcal{R}^{FairKL}$ on Corrupted CIFAR-10 and bFFHQ

|  | Corrupted CIFAR-10 Ratio (%) | | | | bFFHQ Ratio (%) |
|---|---|---|---|---|---|
|  | 0.5 | 1.0 | 2.0 | 5.0 | 0.5 |
| FairKL (mean) | $32.37_{\pm1.72}$ | $35.65_{\pm0.75}$ | $39.94_{\pm0.50}$ | $50.25_{\pm0.16}$ | $60.55_{\pm1.05}$ |
| FairKL (full) | $\mathbf{33.33}_{\pm0.38}$ | $\mathbf{36.53}_{\pm0.38}$ | $\mathbf{41.45}_{\pm0.42}$ | $\mathbf{50.73}_{\pm0.90}$ | $\mathbf{63.70}_{\pm0.90}$ |

## C.5 IMPORTANCE OF THE REGULARIZATION WEIGHT

We conduct an analysis on the importance and stability of the weights $\alpha$ and $\lambda$ of Eq. 11. We perform multiple experiments selecting $\alpha \in \{0.01, 0.1, 1.0\}$. For simplicity, we fix $\epsilon = 0$, and we report the accuracy scored on the Biased-MNIST test. The results are show in Tab. 12. There seems to be a correlation between the value of $\alpha$ and the strength of the bias: for stronger biases it is better to give more importance to the regularization term rather than the target loss function. Additionaly, we also find that $\alpha$ depends on the complexity of the dataset: for example on Corrupted-CIFAR10 and bFFHQ we use $\alpha = 0.1$, for 9-Class ImageNet we use $\alpha = 0.5$.

Table 12: Importance of the weights $\alpha$ and $\lambda$.

| Corr.$\setminus \alpha$ | $\lambda = 0.5$ | | | $\lambda = 1$ | | |
|---|---|---|---|---|---|---|
|  | 0.01 | 0.1 | 1.0 | 0.01 | 0.1 | 1.0 |
| 0.999 | $\mathbf{89.55}_{\pm1.43}$ | $31.63_{\pm2.30}$ | $38.38_{\pm1.26}$ | $\underline{84.98}_{\pm2.29}$ | $43.21_{\pm10.08}$ | $31.84_{\pm4.31}$ |
| 0.997 | $\mathbf{94.08}_{\pm0.10}$ | $82.11_{\pm2.48}$ | $78.91_{\pm2.48}$ | $\underline{91.08}_{\pm0.82}$ | $84.98_{\pm10.23}$ | $79.03_{\pm3.15}$ |
| 0.995 | $\underline{92.42}_{\pm3.76}$ | $90.60_{\pm3.35}$ | $86.63_{\pm2.26}$ | $88.39_{\pm5.00}$ | $\mathbf{93.97}_{\pm1.83}$ | $88.27_{\pm1.72}$ |
| 0.99 | $95.00_{\pm0.21}$ | $\underline{96.60}_{\pm0.17}$ | $93.75_{\pm0.25}$ | $90.72_{\pm0.51}$ | $\mathbf{97.13}_{\pm0.38}$ | $94.74_{\pm0.40}$ |

