# OpenReview forum: "Unbiased Supervised Contrastive Learning"
_ICLR.cc/2023/Conference — ICLR 2023 poster_

### Official Review · Reviewer_BLxm · 2022-10-20

**Confidence:** 4
**Correctness:** 3
**Technical Novelty And Significance:** 3
**Empirical Novelty And Significance:** 2
**Recommendation:** 6

**Clarity, Quality, Novelty And Reproducibility:**

The writing of the paper can be improved. For example, as mentioned above, the related works section should be re-organized and include contrastive learning literature, and the explanation in the method section should be checked and be more clear.

The proposed method is different from existing works, but as mentioned above, it is not enough validated empirically how effective it is. It should be reproducible since most of the details are included in the paper.

**Strength And Weaknesses:**

Pros:
- The topic of this paper is interesting. It discusses whether traditional contrastive losses can deal with biased data, and how to learn a good representation that is robust to biased data.
- This paper proposed a natural and reasonable generalization of the traditional contrastive losses which has more control of the minimal distance between positive and negative samples. Figure 1 presents a nice illustration of how it can mitigate the biased issue.
- The proposed FairKL regularization addresses the biased issue directly, and shows better performance compared with existing methods on biased datasets, including synthesized and real-world datasets.

Cons:
- The related works section is very hard to follow. It just lists a lot of literature on debiasing without organization. It does not discuss how those works are related or different from this paper either. The literature of contrastive learning is totally missing.
- The empirical validation of \epsilon-SupInfoNCE is not convincing. The SimCLR and Max-Margin results in Table 2 are collected from [Khosla et al. (2020)]. However, the results of CE and SupCon are re-implemented and are worse than that reported in [Khosla et al. (2020)]. (The results of SupCon reported in the original paper is 76.5 for CIFAR-100, which is even slightly better than the proposed \epsilon-SupInfoNCE.) How could those four columns be put together in a table? Also, the results on ImageNet are not included - the performance of the proposed method on large-scale datasets has not been validated.
- The experiments on biased datasets are not enough. It does not investigate how the traditional contrastive losses(e.g. SupCon) performs empirically and how much the proposed method can outperform them on biased datasets, as it discusses analytically in the method section. Moreover, it is not clear how much the improvement on the biased datasets comes from \epsilon-SupInfoNCE and how much comes from FairKL. How well will \epsilon-SupInfoNCE perform without FairKL? How much benefit will FairKL add to methods other than \epsilon-SupInfoNCE?

Question:
- For the FairKL, it is reasonable that considering the standard deviation in addition to the mean value of the distances is helpful. However, I do not think the argument of the standard deviation in 3.1.1 (the discussion for EnD)  is correct. The centroid of samples is not the mean of distances, right? In Figure 1(b), even if the green samples and other-color samples have the same centroid, their average distance to the anchor is not the same - the average distance of the green ones is smaller.  Also, why can we assume the distance distribution follows a normal distribution?


**Summary Of The Paper:**

This paper tackles the problem of learning representation robust to biased data, i.e. data containing easy-to-learn but unrelated features. It first derives \epsilon-SupInfoNCE, which adds control of the minimal distance between positive and negative samples to the traditional contrastive losses. Then it proposes FairKL, a regularization loss minimizing the KL divergence between the distribution of distances to bias-conflicting samples and the distribution of distances to bias-aligned samples. Experiments show \epsilon-SupInfoNCE + FairKL outperforms other debiasing methods on several biased datasets.

**Summary Of The Review:**

This paper investigates an interesting problem - how to learn a good representation that is robust to data bias. It proposes a generalized version of contrastive losses and a regularization loss to address the problem. However, it lacks important experiments and the writing should be improved. Therefore, it still needs a lot of work before it is ready to be published.

---

> ### Author Response · Authors · 2022-11-18
> **Authors reponse to Q3 and Q4**
>
> **[Q3] The experiments on biased datasets are not enough. It does not investigate how the traditional contrastive losses(e.g. SupCon) performs empirically and how much the proposed method can outperform them on biased datasets, as it discusses analytically in the method section. Moreover, it is not clear how much the improvement on the biased datasets comes from ϵ-SupInfoNCE and how much comes from FairKL. How well will ϵ-SupInfoNCE perform without FairKL? How much benefit will FairKL add to methods other than ϵ-SupInfoNCE?**
>
> We investigate the debiasing power of ϵ-SupInfoNCE, ϵ-SupCon and SupCon (without FairKL) in Sec 5.1 on Biased-MNIST (Fig.2), showing that both SupCon (and ϵ-SupCon) and ϵ-SupInfoNCE obtain better results than the CE baseline, with the latter being the best when the data are highly biased.
> Moreover, in Sec. C.2 of the appendix, we show that the non-contrastive constraint of SupCon leads to representations that are more clustered based on the bias attribute, hence less suited for debiasing, and we show that this is mitigated by ϵ-SupInfoNCE as it does not put the same strength as SupCon on the positive (and bias-aligned) samples.
>
> We agree with the Reviewer that it would be interesting to investigate the benefit of FairKL when applied to other losses. We have thus performed additional experiments with CE and SupCon, on Biased-MNIST which are now included in the revised text. We report the main results here:
>
> |Method|0.999|0.997|0.995|0.99|
> |-|-|-|-|-|
> |CE|11.8|62.5|79.5|90.8|
> |ϵ-SupCon|24.36|74.35|84.13|91.12|
> |ϵ-SupInfoNCE|33.16|73.86|83.65|91.18|
> ||
> |CE+FairKL| 79.9 | 93.86 | 94.85 | 95.92|
> |ϵ-SupCon+FairKL| 89.45 | 95.75 | 96.31 | 96.72 |
> |ϵ-SupInfoNCE+FairKL| 90.51 | 96.19 | 97.00 | 97.86 |
>
>
> It is worth noting that, when adding FairKL, we almost always obtain better results than without it, with the best performance achieved by ϵ-SupInfoNCE. This shows that both ϵ-SupInfoNCE and FairKL help improve the results.
>
> **[Q4]: For the FairKL, it is reasonable that considering the standard deviation in addition to the mean value of the distances is helpful. However, I do not think the argument of the standard deviation in 3.1.1 (the discussion for EnD) is correct. The centroid of samples is not the mean of distances, right? In Figure 1(b), even if the green samples and other-color samples have the same centroid, their average distance to the anchor is not the same - the average distance of the green ones is smaller. Also, why can we assume the distance distribution follows a normal distribution?**
>
>
> The Reviewer is right, and the situation depicted in Fig.1b does not actually reflect the consideration of Sec. 3.1.1 (some samples were missing in order to obtain the same average distance between bias-aligned and bias-conflicting samples). We have added a new figure (Fig. 3 and 4 in the Appendix) which now represents the correct suboptimal case, and improved the explanation in Sec. 3.1.1. We thank the Reviewer for pointing this out.
>
> We make the assumption of normal distribution for practical reasons (for easily computing the regularization term) and because it works well in our experiments (note that, in practice, one can also optimize the Jeffreys divergence $D_{KL}(p||q)+D_{KL}(q||p)$, or a simplified version such as the difference of the two statistics $(\mu_{+,b} - \mu_{+,b'})^2 + (\sigma_{+,b} - \sigma_{+,b'})^2$).
> We have also tried other formulations, such as for the Log-Normal distribution, but, at least in our experiments, we did not obtain better results. Nevertheless, in other applications or data-sets, other distributions or loss variants could perform better. We have updated the main text to include this point.

---

> > ### Comment · Reviewer_BLxm · 2022-11-28
> > **Follow-up response**
> >
> > Thanks for adding the experiments and revising the incorrect example. They have addressed my concerns of Q3 and Q4.

---

> ### Author Response · Authors · 2022-11-18
> **Authors response to Q1 and Q2**
>
> We thank the Reviewer for the feedback. Here is our response.
>
> **[Q1] The related works section is very hard to follow. It just lists a lot of literature on debiasing without organization. It does not discuss how those works are related or different from this paper either. The literature of contrastive learning is totally missing.**
>
> We thank the Reviewer for the feedback, and we agree that the related works section should be improved. Hence, in Sec. 2 we now include also works on contrastive learning (which were previously mentioned in Sec. 3), and we improved the discussion on debiasing works (we left the detailed comparison with other debiasing techniques in Sec. 3.1.1, after introducing our method, and the technical details.).
>
> **[Q2] The empirical validation of ϵ-SupInfoNCE is not convincing. The SimCLR and Max-Margin results in Table 2 are collected from [Khosla et al. (2020)]. However, the results of CE and SupCon are re-implemented and are worse than that reported in [Khosla et al. (2020)]. (The results of SupCon reported in the original paper is 76.5 for CIFAR-100, which is even slightly better than the proposed ϵ-SupInfoNCE.) How could those four columns be put together in a table? Also, the results on ImageNet are not included - the performance of the proposed method on large-scale datasets has not been validated.**
>
> For CIFAR-10 and 100, in order to employ the original batch size of 1024, we were forced to use mixed precision in our trainings of SupCon, CE, and ϵ-SupInfoNCE due to computing resources limitations. We suppose that this is the reason for the observed difference of ~1% w.r.t. the results presented in [Khosla et al. (2020)]. Please note that for these results we used the official code provided at [https://github.com/HobbitLong/SupContrast](). For the other columns (SimCLR and Max-Margin), we report the original results (not with mixed precision) because the gap in terms of accuracy with respect to Supcon and ϵ-SupInfoNCE is already greater than 2%, and employing mixed precision would have probably resulted in even worse results for SimCLR and Max-Margin.
> We agree with the Reviewer that this was not properly explained in the text, and it has now been fixed in Tab. 2.
>
> For the full ImageNet-1k, as stated in the main text, we were not able to evaluate this dataset due to a lack of adequate computing resources. Reproducing the exact setup as in [Khosla et al. (2020)] e.g. with batch sizes > 6000 requires an amount of VRAM in the order of TBs, such as with Google TPUs, while we are limited to normal GPUs. While we agree that testing large-scale datasets would be an important contribution and that testing extremely resource-heavy setups could give interesting insights, we also believe that the investment in massive computing resources and (worth noting) the resulting CO2 impact, make these experiments less appealing in the scope of this paper.

---

> > ### Comment · Reviewer_BLxm · 2022-11-28
> > **Follow-up response**
> >
> > Thank the authors for editing the related work section, but it would be better if the literature on Debiasing can be better-organized. It is too long now. You may want to separate it into different paragraphs or make it more concise.
> >
> > Thanks for the explanation about the empirical results, but it would be better if you reproduce SimCLR and Max-Margin with mixed precision, which can make Table 2 more consistent and informative. Also, it would be good to clarify whether code will be open-sourced upon publication so that others can reproduce the results.
> >
> > Given the current revised submission, I am happy to raise my score if the authors would like to **fix Table 2** and **fully open-source** the code.

---

> > > ### Author Response · Authors · 2022-12-02
> > > **Authors follow-up response**
> > >
> > > We thank the Reviewer for the suggestion about the related work section. We will address it in the final version of the paper, if accepted, by separating the "Debiasing" part into paragraphs and reducing the "ensembling approaches" paragraph, to make it easier to read.
> > >
> > > As per reviewer request, we have also run the experiments with the SimCLR baseline using the official code available at https://github.com/HobbitLong/SupContrast with mixed precision, obtaining these results (full results with std will be inserted in the revised text):
> > >
> > > |Dataset|Network|Max-Margin|SimCLR (Khosla)| SimCLR*|CE*|SupCon*|ϵ-SupInfoNCE*|
> > > |-|-|-|-|-|-|-|-|
> > > |CIFAR-10|ResNet-50|92.4|93.6|91.74|94.73|95.64|96.14|
> > > |CIFAR-100|ResNet-50|70.5|70.7|68.94|73.43|75.41|76.04|
> > > |ImageNet-100|ResNet-50|-|-|66.14|82.1|81.99|83.3|
> > >
> > > *denotes results with mixed precision. For both CIFAR-100 and CIFAR-100 we observe, as expected, a small margin of ~ 1-2% with the reported results in [Khosla et al. (2020)].
> > > The max-margin implementation is not available in the above repository, and an official code by [Liu et al. (2016)] is only provided in Caffe, which does not support automatic mixed precision. Additionally, we also tested SimCLR on ImageNet-100 with the setup described in B.1.2, although we do not have a reference value in [Khosla et al. (2020)] for this dataset. We will update the results in the revised text accordingly.
> > >
> > > The source code of this work is available in the supplementary zip archive, and an updated version will be publicly released on github upon acceptance.

---

> > > > ### Comment · Reviewer_BLxm · 2022-12-02
> > > > **Re: Authors follow-up response**
> > > >
> > > > Thanks for the responses. They addressed my concerns and I am happy to raise the rating.

---

### Official Review · Reviewer_pTc7 · 2022-10-22

**Confidence:** 3
**Correctness:** 4
**Technical Novelty And Significance:** 3
**Empirical Novelty And Significance:** 3
**Recommendation:** 8

**Clarity, Quality, Novelty And Reproducibility:**

The manuscript is clear and of high quality. I am not an expert in this area, but in trusting the authors that the derivations of related methods + eps-SupInfoNCE are original, then the work is sufficiently novel. The methods are well explained, and I don't see reproducibility as being an issue, but at the same time, the authors seem to not be releasing code, which is unfortunate.

**Strength And Weaknesses:**

The paper is very clear, especially when taking into account the derivations in the appendix that would usually be glossed over. I liked that the authors include both a thorough related work section *and* section 3.1.1, which makes direct comparisons with closely related work after introducing the method (for a more in-depth understanding). In addition to formulating eps-SupInfoNCE and eps-SupCon, the discussion around prior work, particularly Equations 6 and 7, is interesting and insightful. Calling Equation 2 a 'theoretical framework' might be a bit much, but still it is insightful, both in the context of prior work and the proposed methods. The experiments do admittedly lie on the toy-ish side, but are for the most part convincing.

I see only one potential weakness: it appears to me that FairKL is applicable to many constrastive learning methods; is this correct or incorrect? If correct, then there is an experimental gap, in that, with respect to handling bias, the comparison is always between eps-SupInfoNCE + FairKL vs. other methods. Why are they intertwined in the experiments? To evaluate FairKL, wouldn't it make sense to include many experiments without eps-SupInfoNCE?

**Summary Of The Paper:**

The authors use a common approximation of the max operator to simply motivate and formulate eps-SupInfoNCE, their proposed loss for contrastive learning, and FairKL, their proposed regularization term for mitigating effects of bias in datasets. Also using this approximation, they formulate prior work such as InfoNCE, InfoL1O, and SupCon, and highlight some of their deficiencies. They demonstrate eps-SupInfoNCE and FairKL across a variety of common vision datasets and their biased variants, showing that eps-SupInfoNCE performs favorably to alternatives on CIFAR-10, CIFAR-100, and a small variant of ImageNet, and showing that the combination of eps-SupInfoNCE + FairKL performs favorably on Biased-MNIST, Corrupted CIFAR-10, bFFHQ (faces of younger females and older males), and small variants of ImageNet with known biases.

**Summary Of The Review:**

The 'Summary of the Paper' and 'Strengths and Weaknesses' sections speak for themselves: overall, the simple derivation of related work + new methods is easy to follow and leads to convincing results across various datasets.

---

> ### Author Response · Authors · 2022-11-18
> **Authors response**
>
> We thank the Reviewer for the positive feedback. Here is our response.
>
> **[Q1]: The paper is very clear, especially when taking into account the derivations in the appendix that would usually be glossed over. I liked that the authors include both a thorough related work section and section 3.1.1, which makes direct comparisons with closely related work after introducing the method (for a more in-depth understanding). In addition to formulating eps-SupInfoNCE and eps-SupCon, the discussion around prior work, particularly Equations 6 and 7, is interesting and insightful. Calling Equation 2 a 'theoretical framework' might be a bit much, but still it is insightful, both in the context of prior work and the proposed methods. The experiments do admittedly lie on the toy-ish side, but are for the most part convincing.**
>
> We are glad that the Reviewer appreciates our comparison with the related works. As a minor note, with "theoretical framework" we refer more to the entire process of deriving a differentiable loss function from a starting metric condition (such as Eq.2), which allows the proposed analysis of the different losses.
>
> **I see only one potential weakness: it appears to me that FairKL is applicable to many constrastive learning methods; is this correct or incorrect? If correct, then there is an experimental gap, in that, with respect to handling bias, the comparison is always between eps-SupInfoNCE + FairKL vs. other methods. Why are they intertwined in the experiments? To evaluate FairKL, wouldn't it make sense to include many experiments without eps-SupInfoNCE?**
>
> The Reviewer is correct and FairKL can indeed be applied with different loss functions. We reported results with ϵ-SupInfoNCE as it performs better than SupCon or ϵ-SupCon. However, in light of the feedback received, we also included an additional ablation study (on Biased-MNIST), now included in the revised text, with different losses. The results are summarized below:
>
> |Method|0.999|0.997|0.995|0.99|
> |-|-|-|-|-|
> |CE+FairKL| 79.9 | 93.86 | 94.85 | 95.92|
> |ϵ-SupCon+FairKL| 89.45 | 95.75 | 96.31 | 96.72 |
> |ϵ-SupInfoNCE+FairKL| 90.51 | 96.19 | 97.00 | 97.86 |
>
> It is worth noting that with FairKL we obtain better results than most of the other baselines (in the full Tab.2) with either CE, ϵ-SupCon or ϵ-SupInfoNCE; with the best performance achieved by ϵ-SupInfoNCE.
>
>
> **[Q2]: The methods are well explained, and I don't see reproducibility as being an issue, but at the same time, the authors seem to not be releasing code, which is unfortunate.**
>
> We would like to point out that the code is attached in the supplementary zip archive. However, the Reviewer is right as we did not mention this in the text. We apologize for the lack of clarity.

---

### Official Review · Reviewer_DYGt · 2022-10-24

**Confidence:** 3
**Correctness:** 3
**Technical Novelty And Significance:** 2
**Empirical Novelty And Significance:** 3
**Recommendation:** 6

**Clarity, Quality, Novelty And Reproducibility:**

Clarity:
The proposed method is interesting, however, it would be better to give more intuitive explanations about why the reformulations of the contrastive losses can properly address the biases in datasets. In particular, Section 3.1 could be improved with some visual illustration of the loss design.

Quality:
The paper contains solid formulations and experiments with comparison on different datasets. More efforts could be spent to make the method formulation clearer.

Novelty:
The proposed method is related to a couple of existing methods as indicated in Section 3.1.1, but it also has constraints different from other methods.

Reproducibility:
The authors provide in the supplementary for reproducing their results.



**Strength And Weaknesses:**

+ The paper studies an important problem and proposes reformulations upon the contrastive loss to address the biases in the data.

- The motivation that drive the model formulation is not clear. In particular, t is not very clear why contrastive losses should be used as for the formulations to address the biases in the data. Why can't one also consider other metric loss such as the triplet loss, or consider classification loss that also allow to introduce margin or temperature to address the biases in the data?

- The proposed method lacks intuitive explanations about why the reformulations of contrastive loss can properly address the biases in the data. From the formulations, it turns out the authors introduce an additional margin parameter to ensure the relevant distance of a positive wrt an anchor and the nearest negative. The formulations are quite similar to the idea of triplet loss  [a,b], but it is unclear why and how it resolve biases in data.

- What kinds of biases do the paper aims to tackle? It seems that the bias in this paper is quite related to domain drift across datasets. How does the proposed technique compared to ones that address the domain drift in data?

- There is an existing formulation of supervised contrastive loss proposed in [c]. What is the difference between the SupInfoNCE proposed in this paper and the one proposed in [c]?

[a] FaceNet: A Unified Embedding for Face Recognition and Clustering. CVPR 2015
[b] In Defense of the Triplet Loss for Person Re-Identification. arXiv2017
[c] Supervised Contrastive Learning. NeuRIPS 2020



**Summary Of The Paper:**

The paper tackles the problem of learning representations that are robust to biases in the data. The authors first clarify why existing contrastive losses fail to deal with biased data, and further derive a novel formulation of supervised contrastive loss to provide more accurate control of minimal distance between positive and negative samples. Moreover,the authors propose a new debiasing regularization loss to deal with extremely biased data.

To evaluation the proposed losses, the authors first provide a benchmark on standard classification datasets CIFAR10, CIFAR100, ImageNet to evaluate different formulations of the proposed loss terms. The authors further conduct experiments on biased datasets Biased-MNIST, Corrupted CIFAR-10, bFFHQ and 9-Class ImageNet and ImageNet-A, which contains color biases or texture biases in the data. The experiments show better results in comparison to other debiasing techniques

**Summary Of The Review:**

The paper studies an important problem and proposes reformulations upon the contrastive loss. Experiments are conducted to verify the effectiveness of the proposed method. However, the writing about proposed method is suggested to be improved with better clarity as mentioned above in the section of Strength And Weaknesses.

---

> ### Author Response · Authors · 2022-11-18
> **Authors response to Q3 and Q4**
>
> **[Q3]: What kinds of biases do the paper aims to tackle? It seems that the bias in this paper is quite related to domain drift across datasets. How does the proposed technique compared to ones that address the domain drift in data?**
>
> Our work mainly deals with selection biases and domain generalization issues. These biases usually result in training data that does not accurately reflect the true real-world distribution. For example, in the bFFHQ dataset for age prediction, the samples are biased with respect to the gender. In this case, the model learns to focus on "peripheral" attributes ($B$ = gender) correlated to the target features ($Y$ = age), which yields high accuracy on the original domain (training set), but it fails when the test distribution drifts to a different correlation between $Y$ and $B$.
> Another example related to domain adaptation/drift is represented by the ImageNet-A experiment: this dataset contains samples in the test set that are harder to be correctly classified since the subjects in the images may appear in different contexts and scenes than in the training set (as explained in [Beery et al. (2018)]).
> As pointed out by the Reviewer, issues such as selection biases and domain adaptation are very related, and, indeed, many of the techniques we compared to can also be applied to domain adaptation/generalization tasks, as shown in [Wang et al. (2019); Sagawa et al. (2020)].
>
> *Beery, S., *et al*. Recognition in terra incognita. ECCV 2018.; Wang, H. et al. Learning Robust Representations by Projecting Superficial Statistics Out, ICLR 2019; Sagawa, S. et al. Distributionally Robust Neural Networks, ICLR 2020*.
>
> **[Q4]: There is an existing formulation of supervised contrastive loss proposed in [c]. What is the difference between the SupInfoNCE proposed in this paper and the one proposed in [c]?**
>
> The differences between ϵ-SupInfoNCE and SupCon ([c]) are discussed in Sec.3 and are:
> * SupCon lacks the margin ϵ in its formulation,
> * in contains an additional non-contrastive condition (over the positive samples)
>
> |Condition|Loss|
> |-|-|
> |$s^-_j - s^+_i \leq 0 \quad \forall i,j \,\text{ and }\, s^+_t - s^+_i \leq 0 \quad \forall i,t \neq i$|SupCon|
> |$s^-_j - s^+_i \leq - \epsilon$|ϵ-SupInfoNCE|
>
> And we also introduce ϵ-SupCon, which contains ϵ in the starting condition.
> However we show that the non-contrastive condition $s^+_t - s^+_i \leq 0$ can actually be harmful when dealing with biased data, as it increases the importance of positive and bias-aligned samples (the majority), which lead to biased representations. ϵ-SupInfoNCE, on the other hand, is probably the most straightforward extension of InfoNCE to the supervised case, and can also perform better than SupCon on biased data (due to the lack of the non-contrastive condition and the presence of ϵ).
>
> *[1] Kihyuk Sohn. Improved Deep Metric Learning with Multi-class N-pair Loss Objective. NeurIps. 2016, [a] FaceNet: A Unified Embedding for Face Recognition and Clustering. CVPR 2015 [b] In Defense of the Triplet Loss for Person Re-Identification. arXiv2017 [c] Supervised Contrastive Learning. NeuRIPS 2020*

---

> ### Author Response · Authors · 2022-11-18
> **Authors response to Q1 and Q2**
>
> We thank the Reviewer for the feedback. Here is our response.
>
> **[Q1+Q2]: The motivation that drive the model formulation is not clear. In particular, t is not very clear why contrastive losses should be used as for the formulations to address the biases in the data. Why can't one also consider other metric loss such as the triplet loss, or consider classification loss that also allow to introduce margin or temperature to address the biases in the data?**
>
> **[Q1+Q2]: The proposed method lacks intuitive explanations about why the reformulations of contrastive loss can properly address the biases in the data. From the formulations, it turns out the authors introduce an additional margin parameter to ensure the relevant distance of a positive wrt an anchor and the nearest negative. The formulations are quite similar to the idea of triplet loss [a,b], but it is unclear why and how it resolve biases in data.**
>
> The use of a metric learning approach within a contrastive learning framework allows us to easily model and remove the bias effect from the data. We believe that using distances within a metric representation space to characterise positive, negative, bias-aligned and bias-conflicting samples is a simple and highly interpretable way and allows us to easily add justifiable constraints and regularizations. Furthermore, lately, contrastive learning approaches have reached SOTA results in many vision applications. This is why we have chosen this kind of approach instead of other possible approaches (e.g., classification losses). Please note that we have also confirmed the superiority of contrastive learning approaches in this paper (See Tab.2)
>
> Furthermore, the Reviewer is correct and we could indeed use any contrastive/metric learning loss [1], such as contrastive losses (pairs of data coming from the same class), triplet loss (1 positive and 1 negative), InfoNCE or N-Pair loss (1 positive and multiple negatives) or SupCon,ϵ-SupCon,ϵ-SupInfoNCE (multiple positive and multiple negatives). Here, we have focused on the most recent losses, since they have been shown to produce the best results in vision applications. Moreover, we would like to specify that throughout the paper, when using the term "contrastive losses", we refer to any contrastive learning loss and not necessarily to loss based on pairs of examples as in [1]. We have clarified this point in the Introduction.
>
> Regarding the margin, we have shown that by increasing it, we can achieve a better separation between positive and negative samples and, especially in the highly biased settings, it can also mitigate the effect of the bias (See Table 3 and Fig. 1). Interestingly, this is true independently on the used loss (e.g., SupCon or SupInfoNCE).
>
> Even if the margin plays an importan role, in many situations it can only mitigate the effect of the bias and not remove it. This is why we have also proposed FairKL, which has been imagined to fully remove the effect of the bias. By leveraging the proposed metric learning approach, we analysed the related debiasing methods, such as EnD, understood and described their limitations and proposed a new set of contraints, FairKL, with a theoretical higher debiasig power. Moreover, the use of FairKL is not only limited to the proposed ϵ-SupInfoNCE (or to contrastive learning losses), even if we empirically found that it gives the best results. In fact, in light of the received feedbacks, we performed additional experiments (on Biased-MNIST), applying FairKL also to ϵ-SupCon and to the cross-entropy loss.
> The results are summarized below:
>
> |Method|0.999|0.997|0.995|0.99|
> |-|-|-|-|-|
> |CE+FairKL| 79.9 | 93.86 | 94.85 | 95.92|
> |ϵ-SupCon+FairKL| 89.45 | 95.75 | 96.31 | 96.72 |
> |ϵ-SupInfoNCE+FairKL| 90.51 | 96.19 | 97.00 | 97.86 |
>
> It is interesting to notice that using FairKL we obtain better results than with most of the other baselines reported in Tab. 2 (e.g., CE, ϵ-SupCon or ϵ-SupInfoNCE).
> The text has been revised accordingly.
>
> Please note that we could also slightly modify the proposed starting metric condition to obtain:
>
> $a) s^- - s^+ \leq 0 \quad \quad b)s^-_j - s^+  \leq 0 \quad \forall j \quad \quad c) s^-_j - s^+_i \leq -\epsilon \quad \forall j,i$
>
> We have that a) corresponds to the triplet loss, b) to the N-Pair loss and c) to our ϵ-SupInfoNCE.
>
> In [Sohn et al. (2016)], it has been shown that N-Pair loss may work better than the triplet-loss. Our loss can be seen as a futher extension to multiple positives and with a controllable margin (c).

---

### Author Response · Authors · 2022-11-18
**Authors general response**

We thank the Reviewers for the thoughtful and useful feedback. We have provided our response to each of the points raised.
Based on the received reviews, we have updated our manuscript (changes highlighted in blue).

---

### Decision · Program_Chairs · 2023-01-20

**Decision:**

Accept: poster

**Justification For Why Not Higher Score:**

The paper gives an interesting, and well-justified, but somewhat narrow modification to the existing supervised contrastive learning framework. A spotlight would need more algorithmic novelty.

**Justification For Why Not Lower Score:**

The paper studies a really important (but, relatively under-studied) issue of deceptively over-predictive features in training. It proposed solution is well-justified and widely applicable. The paper deserves to be published.

**Metareview: Summary, Strengths And Weaknesses:**

Summary:
This paper considers the problem where certain features are extremely predictive in the training data but not predictive in the test data. They first provide a quantitative intuition of why this is a problem, and then propose a modification of supervised contrastive learning to fix this issue.


Strengths:
1. Clearly written paper on a common and important problem.
2. Good quantitative intuition motivating their approach, ablation studies and experiments
3. Good survey and contrast to existing work
4. Detailed replies and improvements to the paper in the discuss period

Weaknesses:

Better evaluation of existing baselines was mentioned by a couple of the reviewers

**Note From Pc:**

if the above contains the word "oral" or "spotlight" please see: "oral" presentation means -> notable-top-5% and "spotlight" means -> notable-top-25%. As stated in our emails, we are disassociating presentation type from AC recommendations